



# From fine to giant: Multi-instrument assessment of the dust particle size distribution at an emission source during the J-WADI field campaign

Hannah Meyer[1], Konrad Kandler[2], Sylvain Dupont[3], Jerónimo Escribano[4], Jessica Girdwood[5,6,7], George Nikolich[8], Andrés Alastuey[9], Vicken Etyemezian[8], Cristina González-Flórez[4,10], Adolfo González-Romero[4,9], Tareq Hussein[11,12], Mark Irvine[3], Peter Knippertz[1], Ottmar Möhler[13], Xavier Querol[9], Chris Stopford[5], Franziska Vogel[13,14], Frederik Weis[15], Andreas Wieser[1], Carlos Pérez García-Pando[4,16], and Martina Klose[1]

[1]Karlsruhe Institute of Technology (KIT), Institute of Meteorology and Climate Research Troposphere Research (IMKTRO), Karlsruhe, Germany.
[2]Technical University of Darmstadt, Germany.
[3]INRAE, Bordeaux Sciences Agro, ISPA, Villenave d'Ornon, France.
[4]Barcelona Supercomputing Center (BSC), Barcelona, Spain.
[5]Particle Instruments and Diagnostics, University of Hertfordshire, Hatfield, AL109AB.
[6]Now at: Centre for Atmospheric Science, School of Earth and Environmental Science, University of Manchester, Manchester, M13 9PL.
[7]Now at: National Centre for Atmospheric Science, School of Earth and Environmental Science, University of Manchester, Manchester, M13 9PL.
[8]Desert Research Institute (DRI), NV, USA.
[9]Institute of Environmental Assessment and Water Research - Consejo Superior de Investigaciones Científicas (IDAEA-CSIC), Spain.
[10]Now at: Danish Meteorological Institute (DMI), Copenhagen, Denmark.
[11]Environmental and Atmospheric Research Laboratory (EARL), Department of Physics, School of Science, University of Jordan, Amman 11942, Jordan.
[12]Institute for Atmospheric and Earth System Research (INAR/Physics), Faculty of Science, University of Helsinki, PL 64, FI-00014 UHEL, Helsinki, Finland.
[13]Karlsruhe Institute of Technology (KIT), Institute of Meteorology and Climate Research Atmospheric Aerosol Research (IMKAAF), Germany.
[14]Now at: Institute of Atmospheric Sciences and Climate (ISAC), National Research Council (CNR), Bologna, Italy.
[15]Palas GmbH, Germany.
[16]Catalan Institution for Research and Advanced Studies (ICREA), Spain.

**Correspondence:** Hannah Meyer (hannah.meyer@kit.edu)

**Abstract.**

Mineral dust particles emitted from dry, uncovered soil can be transported over vast distances, thereby influencing climate and environment. Its impacts are highly size-dependent, yet large particles with diameters $d_p > 10\,\mu m$ remain understudied due to their low number concentrations and instrumental limitations. Accurately characterizing the particle size distribution

5 (PSD) at emission is crucial for understanding dust transport and climate interactions.





Here we characterize the dust PSD at an emission source during the Jordan Wind Erosion and Dust Investigation (J-WADI) campaign, conducted in Wadi Rum, Jordan, in September 2022, focusing on super-coarse ($10 < d_p \leq 62.5\,\mu m$) and giant ($d_p > 62.5\,\mu m$) particles. This study is the first to continuously cover the full range of diameters from $d_p = 0.4$ to $200\,\mu m$ at an emission source by using a suite of aerosol spectrometers with overlapping size ranges. This overlap enabled a systematic intercomparison and validation across instruments, improving PSD reliability.

Results show significant PSD variability over the course of the campaign. During periods with friction velocities ($u_{*t}$) above $0.22\,\mathrm{m\,s^{-1}}$, the approximate onset of local dust emissions, super-coarse and giant particles were observed, with concentrations increasing with $u_*$. These large particles accounted for about 90% of the total mass concentration during the campaign. A prominent mass concentration peak was observed near $d_p = 60\,\mu m$ in geometric diameter. While particle concentrations for $d_p < 10\,\mu m$ showed good agreement among most instruments, discrepancies appeared for larger $d_p$ due to reduced instrument sensitivity at the size range boundaries and sampling inefficiencies. Despite these challenges, physical samples collected using a flat-plate sampler largely confirmed the PSDs derived from the aerosol spectrometers. These findings help to advance our understanding of the dust PSD and the abundance of super-coarse and giant particle at emission sources.

# 1 Introduction

Mineral dust aerosol originates from the suspension of particles of uncovered dry soil under conditions of strong enough winds. It represents the dominant fraction of the global aerosol mass in Earth's atmosphere (Textor et al., 2006), with a significant impact – from emission to deposition – on atmospheric processes and climate dynamics (e.g., Shao et al., 2011; Kok et al., 2023). Mineral dust affects Earth's energy balance by scattering and absorbing solar and infrared radiation (e.g., Ryder et al., 2013; Kok et al., 2018; Di Biagio et al., 2020) and influencing cloud formation and precipitation potential (e.g., Kumar et al., 2011; Hoose and Möhler, 2012; Froyd et al., 2022). It also transports nutrients to ecosystems, impacting carbon uptake and atmospheric $CO_2$ levels (Goudie and Middleton, 2001; Jickells et al., 2005; Schulz et al., 2012). Its overall radiative forcing remains highly uncertain, estimated at $-0.2\pm0.5\,\mathrm{W\,m^{-2}}$, leaving it unclear whether dust ultimately warms or cools the climate (Kok et al., 2023).

The climate impact of dust aerosol is not only determined by the amount, shape, and mineralogical composition of the particles, but also by their particle size distribution (PSD, Mahowald et al., 2014). The optical diameters ($d_o$) typically used to report the PSDs obtained with optical particle counters (OPCs) are diameters of non-absorbing reference particles that produce the same scattered light intensity as the measured dust particles. Instead, dust modeling typically uses the geometric or volume-equivalent diameter ($d_{geo}$), which refers to the diameter of a sphere with the same volume as the aspherical particle. Area-equivalent (or projected area) diameter ($d_{\mathrm{PA}}$) measures the diameter of a circle with the same 2D projected area as the dust particle (Kandler et al., 2007). For dust, the PSD is divided into different size ranges: fine dust with a diameter smaller than $d_p < 2.5\,\mu m$, coarse dust with $2.5 \leq d_p < 10\,\mu m$, super-coarse dust with $10 \leq d_p < 62.5\,\mu m$, and giant dust with $d_p \geq 62.5\,\mu m$ (Adebiyi et al., 2023).



Coarse, super-coarse, and giant dust particles tend to warm the atmosphere because their absorption increases more significantly than their scattering as particle size grows (Adebiyi et al., 2023). This is reflected in their lower single-scattering albedo

(SSA; ratio of scattered to total extinguished radiation), which decreases from $\sim 0.80$ at $d_p = 10\,\mu m$ to even lower values for larger particles (Tegen et al., 1996; Adebiyi et al., 2023). In addition to absorbing shortwave radiation, these larger particles are also strong absorbers of longwave radiation, further contributing to atmospheric warming (Tegen et al., 1996; Dufresne et al., 2002). In contrast, smaller dust particles primarily cool the atmosphere by efficiently scattering solar radiation, with an SSA close to 1 that decreases to approximately 0.95 at $d_p = 2\,\mu m$ (Kok et al., 2023). Thus, smaller particles dominate cooling

effects, while larger particles contribute to atmospheric warming (Kok et al., 2017; Ryder et al., 2019; Kok et al., 2023).

Particle size is also important for cloud microphysics and precipitation processes (Min et al., 2009). Large dust particles can act as effective cloud condensation nuclei, promoting the formation of very large cloud droplets that enhance collision-coalescence processes, leading to precipitation (Mahowald et al., 2014).

Despite of their importance in dust-climate interactions, coarse to giant dust particles have traditionally been presumed to

sediment rapidly (e.g., Seinfeld and Pandis, 1998), limiting their atmospheric lifetime and potential for long-range transport. Despite expectations of rapid settling, studies have sampled giant dust particles up to $450\,\mu m$ in diameter several thousand kilometers from their source regions (Betzer et al., 1988; van der Does et al., 2018). van der Does et al. (2018) estimated that $100\,\mu m$ particles could travel distances of up to $438\,km$ at wind speeds of $25\,m\,s^{-1}$ from an altitude of $7\,km$, assuming a deposition velocity of $400\,mm\,s^{-1}$. They concluded that this estimate cannot explain the long travel distances observed. In

contrast, the prolonged suspension of giant particles suggests that mechanisms must exist to counteract gravitational settling (e.g., turbulence, convective uplift, or electrostatic forces), yet these processes are not fully understood and remain an active area of research (Rosenberg et al., 2014; van der Does et al., 2018; Harper et al., 2022; Ratcliffe et al., 2024). Since the mechanisms driving the transport and prolonged suspension of large particles are poorly understood, their emission and transport are either excluded from numerical simulations or their presence in the atmosphere is underestimated (Adebiyi and Kok, 2020). As

a result, most models cannot represent the impacts of super-coarse and giant particles on climate, which introduces significant uncertainties into dust-climate impact assessments (Kok et al., 2023).

To better understand the mineral dust cycle and to include its impacts on climate into models, it is essential to accurately quantify and characterize the PSD of mineral dust, including super-coarse and giant particles. Accurately measuring the full PSD of mineral dust remains challenging, as no single instrument can cover its entire size spectrum (Mahowald et al., 2014).

Giant dust particles are especially difficult to measure due to their relatively low expected number concentrations and the low sampling efficiencies of instrument inlets (Adebiyi et al., 2023; Schöberl et al., 2024). Aerosol instruments that actively draw in air are susceptible to sampling biases: Deviations from isokinetic sampling (where the airflow inside the inlet matches the ambient wind speed) can cause over-/underestimation of larger particles due to departures of their trajectories from the flow streamlines. Additionally, inertial losses in tubing and gravitational settling in horizontal sampling lines often lead to under-

sampling of super-coarse and giant particles, distorting the observed PSD (Kulkarni et al., 2011). To mitigate these issues, some aircraft measurement campaigns have avoided inlets altogether or explicitly quantified their losses, allowing for more accurate retrievals of super-coarse and giant dust particles (e.g., Rosenberg et al., 2014; Ryder et al., 2019). These studies give





important insights into the PSD evolution during transport of mineral dust. However, as measured in several hundred meters height, they cannot provide much information about the PSD of dust directly after its emission.

Ground-based measurements at emission sources have predominantly targeted the fine and coarse fractions ($< 10\,\mu$m; Formenti and Di Biagio, 2024), and no campaign has yet comprehensively captured the full size spectrum from fine to giant mineral dust directly at an emission source. In addition to this gap, the variability of the emitted dust PSD is not yet fully understood. For example, some studies predicted that the dust PSD at emission is influenced by wind speed (e.g., Alfaro et al., 1997; Shao, 2004; Ishizuka et al., 2008), others suggested that the dust size distribution is independent of it, at least in the fine

and coarse ranges (Kok, 2011). Shao et al. (2020) and Khalfallah et al. (2020) argued that the dust PSD at emission is influenced by atmospheric boundary-layer stability. Dupont (2022) suggested that friction velocity and air relative humidity may be the primary factors affecting the emitted dust PSD in these previous studies while stability may have no direct influence.

To address the gap in measuring the full PSD of mineral dust, including (super-) coarse and giant particles, and to capture its variability at a desert dust emission source, we conducted field measurements using a comprehensive suite of active and passive

(including open-path) aerosol spectrometers and compared them with flat-plate sampler probes. We addressed key challenges in measuring large dust particles at emission by minimizing the use of inlets, accounting for inlet efficiencies, and resolving inter-instrument uncertainties.

In Sect. 2, we provide a detailed description of the field campaign setup and the instruments used, along with their respective measurement principles and data processing. Section 3 presents the results on the observed dust concentration PSD and its

variability and uncertainties, followed by a comparison of our findings with other studies. Finally, we conclude in Sect. 4 with implications of our results on future research on mineral dust and its climate impacts.

## 2 Methods

The goal of our study is to better quantify the full-range mineral dust concentration PSD and its variability at emission. For this purpose, we use meteorological and aerosol spectrometer measurements collected during the J-WADI field campaign. To

combine measurements from multiple aerosol spectrometer instruments, we apply a strict correction procedure and compare them with collected samples as detailed in the following.

### 2.1 J-WADI field campaign

The J-WADI (Jordan Wind erosion And Dust Investigation) field measurement campaign (https://www.imk-tro.kit.edu/english/11800.php, last accessed 5 Mar. 2025) was conducted in Wadi Rum, Jordan in September 2022. Its aim was to advance our

understanding of the particle size and mineralogical composition of the emitted dust and their dependence on the parent soil and meteorological conditions. J-WADI was co-organized by the ERC Consolidator Grant FRAGMENT (FRontiers in dust minerAloGical coMposition and its Effects upoN climaTe; earlier studies in this context: González-Flórez et al., 2023; González-Romero et al., 2023; Yus-Díez et al., 2023; González-Romero et al., 2024b, a) at the Barcelona Supercomputing





Center (BSC) and the Helmholtz Young Investigator Group "A big unknown in the climate impact of atmospheric aerosol:

Mineral soil dust" at KIT in collaboration with the University of Jordan.

The field site was located at 29°44'21"N, 35°22'56"E, in Wadi Rum (Fig. 1a) and nearby the village of Rashidiyah and downwind of the Quweira solar power plant. It was situated within a flat, open landscape surrounded by small hills (< 100 m in altitude, Fig. 1b). This configuration created a wide opening facing the expected predominant wind direction, while the opposite side featured a narrow opening where winds typically exited. Despite this surrounding topography, the measurement

site was within a flat area that may occasionally be flooded during heavy rainfall periods, and lacked any significant surface roughness features. The location and timing of the campaign were chosen based on analysis of remote sensing data, on-site inspection, and local guidance, considering scientific and practical aspects, such as expected dust emission potential and likelihood, accessibility, and logistics.

The site setup was similar to previous FRAGMENT campaigns in Morocco and Iceland (González-Flórez et al., 2023;

Dupont et al., 2024), but with additional instrumentation emphasizing super-coarse and giant dust and atmospheric turbulence. To minimize the potential for instrument shadowing, we oriented the instruments approximately perpendicular to the expected predominant wind direction (Fig. 1b, c, d), determined by analysis of measurements at seven stations across Jordan available through the NOAA ISD meteo data (https://www.ncei.noaa.gov/products/land-based-station/integrated-surface-database, last accessed 24 Mar. 2025), and observation of local erosion, e.g. ripples. Here we present only instruments and data used in this

study. Other measurements from J-WADI are described elsewhere (e.g., Dupont et al., 2024).

### 2.1.1   Meteorological retrievals

We obtained the friction velocity ($u_*$) and Obukhov length ($L$) using a scintillometer (Scintec SLS-40, 1 min data rate) positioned parallel to the main instrument line (Fig. 2 a). The primary reason for this choice is that the scintillometer provides values representative of a larger surface area, reducing local turbulence biases. The setup of the scintillometer consists of two

primary components, both placed at $z = 2.54\,\mathrm{m}$ height: a transmitter, which emits a laser beam, and a receiver (located $97\,\mathrm{m}$ away from the transmitter), which captured the transmitted light (Fig. 1c). Variations in air temperature along the path of the laser beam cause fluctuations in the intensity of the received light. Detecting these fluctuations gives information about turbulence along the beam's trajectory. Atmospheric stability classes were determined using $z/L$ intervals following the classification by Berg et al. (2011). The classification distinguishes five stability regimes: unstable ($z/L \leq -0.2$), near-unstable

($-0.2 < z/L < -0.04$), neutral ($-0.04 \leq z/L \leq 0.04$), near-stable ($0.04 < z/L < 0.2$), and stable conditions ($z/L \geq 0.2$).

Some data gaps occurred due to scintillometer issues, including high noise levels, power cuts, and overheating. Gaps in $u_*$ and $L$ were filled based on measurements of a 3D sonic anemometer (Campbell Scientific® CSAT3, 50 Hz, Fig. 2b) mounted on a $10\,\mathrm{m}$ tower at $3.0\,\mathrm{m}$ height retrieved based on the eddy covariance (EC) method, as presented in Dupont et al. (2024). To calculate particle concentrations from particle counts detected by the UCASS and SANTRI2 instruments (described in

Sect. 2.1.2 and mounted on a rotating mast), we used wind speed data at $2\,\mathrm{m}$ and $4\,\mathrm{m}$ height measured by 3D sonic anemometers (R. M. Young Company, model 81000 Ultrasonic Anemometer, 40 Hz, Fig. 2d) mounted on a $4\,\mathrm{m}$ mast located less than $2\,\mathrm{m}$ from the mobile mast (Fig. 1c). Pressure was measured using a barometer (R.M. Young Company, Model 61202V) posi-





**Figure 1.** Field location and set-up. (a) Background image from Bing maps with the field site marked with a pink cross, (b) topographic map, field set-up, and surrounding of J-WADI, (c) set-up at the measurement site with the instruments used marked in pink (SCINTTRANS/-REC = scintillometer transmitter/receiver, ROTMAST = rotating mast, DEP = deposition sampler), (d) photo of the site center including most instruments used in this study in their field deployment. Background map copyright (a) © Microsoft, (b) and (c) Esri, Maxar, Earthstar Geographics, and the GIS User Community.





**Figure 2.** Meteorological instruments deployed during J-WADI and used in this study: (a) Scintillometer SLS40 used to retrieve the friction velocity $u_*$ with transmitter in the foreground and receiver in the background. (b) 3D sonic anemometer Campbell Scientific® CSAT3B used to fill gaps from $u_*$, $L$ and wind speed. (c) temperature, humidity and GPS sensor mounted on a 4 m meteorological mast. (d) 3D sonic anemometer R. M. Young Company, model 81000, mounted on the 4 m mast and used to retrieve particle concentrations from UCASS and SANTRI2.

tioned at a height of 1 m on the 4 m mast, while (potential) temperature and relative humidity were measured (and inferred) using a temperature and humidity sensor (Rotronic, Model MP100) mounted at a height of 2 m on the 4 m mast (Fig. 2c). Some

gaps in the data of the 4 m mast occurred due to power cuts and faulty data acquisition. To fill any gaps in the wind data, we used measurements of the 3D sonic anemometers from the 10 m tower at 2 m and 4 m heights (Fig. 2b). Gaps in temperature, relative humidity, and pressure measurements were filled using instruments of the same type mounted on an identical mobile mast located approximately 500 m upwind of the expected dominant wind direction (MM1 in Fig. 1b).





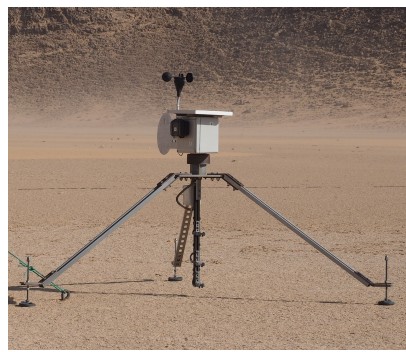

**Figure 3.** SANTRI instrument to measure saltation and retrieve flux density to obtain threshold friction velocity.

**Threshold friction velocity:** The threshold friction velocity $u_{*t}$ represents the minimum friction velocity required to initiate saltation. To retrieve $u_{*t}$, we used saltation data from the SANTRI (Standalone AeoliaN Transport Real-time Instrument) platform (Etyemezian et al., 2017; Goossens et al., 2018; Klose et al., 2019) and implemented the Time Fraction Equivalence Method (TFEM Stout, 1998; Barchyn and Hugenholtz, 2011) for 15-min averaged data over the whole campaign period. This method assumes that the time fraction during which saltation is detected is equivalent to the time fraction during which the friction velocity exceeds the threshold. SANTRI measures saltation counts at three heights as described in González-Flórez et al. (2023, Sect. 2.2.3). Here we counted times of active saltation as those during which at least two out of four sensors measured saltation and the height-dependent streamwise saltation flux (calculated as described in Klose et al. 2019) was non-zero. SANTRI are the original saltation measurement design of which later on the SANTRI2 (Sec. 2.1.2) evolved. It was located approximately 40 m from the tower (Figure 1c, 3). For values of $u_*$, we used the scintillometer data with gaps filled by the 3D sonic anemometer retrieved $u_*$.

### 2.1.2 Aerosol spectrometer measurements

In this study, we analyze a comprehensive set of aerosol spectrometer measurements from the J-WADI campaign to investigate the size distribution of airborne mineral dust particles, with a particular focus on the larger particles (>10 μm). Our aerosol spectrometer suite included the UCASS (Universal Cloud and Aerosol Sounding System, designed at the University of Hertfordshire; Smith et al. 2019), the saltation particle counter SANTRI2 (Standalone AeoliaN Transport Real-time Instrument, second edition, designed at the Desert Research Institute; Etyemezian et al. 2017; Goossens et al. 2018), the CDA (Cloud Droplet Analyzer), the Welas 2500 (White Light Aerosol Spectrometer, Kuhli et al. 2010), and the Fidas 200S (González-Flórez et al. 2023; all three manufactured by the Palas GmbH). This multi-instrument approach was chosen to ensure a robust examination of the entire size distribution from approximately 0.4 to 200 μm, encompassing the fine and giant particle fractions with significant overlap between the size ranges covered by the instruments as shown in Fig. 4. The UCASS and SANTRI2 devices were positioned on the rotating mast (Fig. 5) and the two Welas, two Fidas, and CDA next to or on a scaffolding (Fig. 6). Key instruments properties are summarized in Table 1.





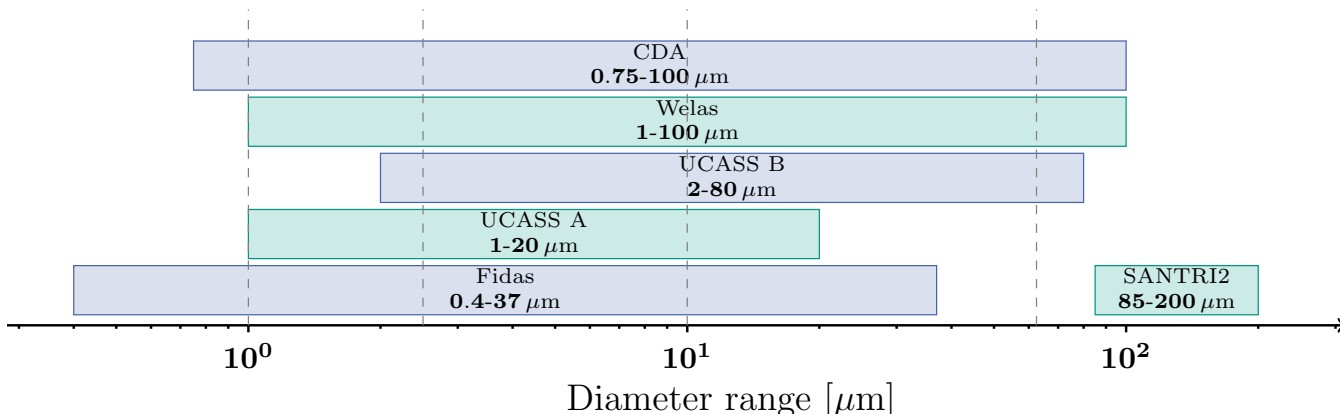

**Figure 4.** Size ranges covered by the five instrument types in terms of optical diameter (except SANTRI2, projected-area diameter). The multi-instrument strategy covers the full size range from about 0.4 to 200 μm, effectively encompassing fine to giant particle sizes. Dashed lines indicate the size ranges of the dust size classifications (Adebiyi et al., 2023).

**Table 1.** Characteristics of the instruments used in the study.

| Instrument | Measurement principle | Light source | Inlet | Diameter size range | Scattering angle | Position | # of bins | Height | Number |
|---|---|---|---|---|---|---|---|---|---|
| UCASS | light scattering | laser: 658 nm | nearly open-path | 1-20 μm, 2-80 μm | 16-104° | rotating mast | 16 | 4 m | 2 |
| Welas | light scattering | Xenon light | directional inlet | 1-100 μm | 90° | scaffolding | 256 | 2/4 m | 2 |
| Fidas | light scattering | LED light | directional inlet | 0.4-37 μm | 90° | scaffolding | 256 | 2/4 m | 2 |
| CDA | light scattering | white light | Sigma-2 head | 0.75-100 μm | 90° | scaffolding | 256 | 4 m | 1 |
| SANTRI2 | shadowing | diode: 890 nm | open-path | 85-200 μm | none | rotating mast | 7 | 2/4 m | 4 (each 5 sensors) |

**SANTRI2:** To explore the giant particle size range, we employed SANTRI2, which uses optical gate devices to infer time and size resolved particle counts (Etyemezian et al., 2017; Goossens et al., 2018; Klose et al., 2019). Originally designed to measure saltation, these instruments are normally positioned vertically on the ground to detect particles transported in saltation at different heights. For the purpose of detecting giant dust particles – the size range of which fits to that of typical saltation particles – we mounted the SANTRI2 at greater height (2 m and 4 m) oriented horizontally to have multiple synchronous measurements at the same height. One unit consists of 5 sensors and each comprises a photosensor that is 9.53 mm away from the diode with 890 nm light wavelength. The onboard electronics interpret the reduction in light signal caused by a sand grain





**Figure 5.** UCASS and SANTRI2 on the rotating mast: (a) The rotating mast (b) with two UCASS and two SANTRI2 at 4 m height and c) SANTRI2 at 2 m height, (d) view from below the UCASS at 4 m height.

or giant dust particle traveling through the beam as voltage drop of a circuit in which the sensor is incorporated when it arrives
at the photosensitive sensor. The SANTRI2 therefore measure projected-area diameters. We employed four SANTRI2 units
measuring in the size range $d_{PA}$ ~85 μm to 200 μm (projected-area diameter) in 7 size bins. As the largest bin, which extends to
diameters $d_{PA} > 200$ μm, has no upper size limit, we did not use it for this study. The lower and upper diameter limits for each
bin vary over time and were determined based on the recorded sensor reference voltage levels. The four SANTRI2 instruments
were mounted on the rotating mast. It consisted of a wind vane and a rotating pole, so that the instruments were turned toward
the wind. Two units were mounted at 2 m height and two at 4 m height with one unit facing upward and one facing downward at





each height. This setup was chosen to avoid possible biases in particle detection due to interference between particle trajectories and flow around the (relatively slim) instruments' bodies. Analysis showed that the downward-facing units exhibited a high level of noise. They were therefore excluded from further analysis. The exact cause of this behavior remains unclear and requires further investigation.

**UCASS:** UCASS is a low-cost particle counter designed at the University of Hertfordshire. It was used for airborne measurements of aerosol and droplet concentrations and size distributions using, e.g., drones and dropsondes in greater heights (Smith et al., 2019; Girdwood et al., 2020, 2022) while we used it ground-based. Here, we deployed the UCASSs at 4 m on the rotating mast together with the SANTRI2 (Fig. 5a). As the measurement principle, it uses wide-angle elastic light scattering with a passive open-geometry system (nearly open-path). The input beam is a 658 nm continuous-wave diode laser, operating at 10

mW. The optical setup includes a laser with a collimator, a cylindrical lens, and a 2 mm aperture. The laser beam is directed into the instrument using a front-silvered mirror positioned at a 45° angle. When particles intersect the laser beam, they scatter light. An elliptical mirror then gathers the light scattered at angles between 16° and 104°, and focuses it onto the detector, where both the pulse height and duration are measured. We used two versions of the UCASS, measuring in 2 different size ranges: one in a larger size range with diameters $d_o$ from 2 - 80 µm (UCASS B) and the other one in a smaller size range with

$d_o$ = 1-20 µm (UCASS A) in 16 different size bins each. To retrieve the (optical) particle diameter, Generalized Lorenz-Mie Theory (GLMT, Gouesbet 2019) is used. GLMT extends the classical Lorenz-Mie framework to account for scattering by particles under non-uniform or partial illumination, such as those exposed to focused or structured beams (e.g., Gaussian or Bessel beams). It adapts the incident field's spatial characteristics, modifies scattering coefficients, and uses numerical integration over the illuminated particle region to accurately model these complex interactions. To address the influence of outliers affecting

the lower size boundary of the first bins, we excluded the first size bin from the analysis for both instruments.

**Welas and Fidas**: The Welas and Fidas systems measure number and size of aerosols through single-particle light scattering detection at a 90° angle. Both instruments use a white light source (Fidas = LED light, Welas = Xenon light) to homogeneously illuminate a T-shaped volume, minimizing optical limitations, border zone, and coincidence errors to illuminate a small, homogeneously lit measurement volume, minimizing optical limitations and ensuring high measurement accuracy. As particles

traverse this volume, they scatter light pulses of varying intensities, which are detected and analyzed based on Mie theory, assuming spherical particles. To measure a wide size range within the same air volume, the active instruments Welas 2500 and Fidas 200S were connected through an optical tube, allowing both instruments to sense particles in the same air flow (Fig. 6 b). The Fidas 200S measured particles in the size range of 0.4–40 µm, whereas the Welas measured in the range between 1 and 100 µm, extending the joint size range of both instruments to include larger particles. Importantly, the pump of the Welas was

not used; instead, the pump of the Fidas provided a steady flow rate of 4.8 l min$^{-1}$, ensuring consistent sampling conditions for both instruments. This setup, deployed for the first time in a field campaign, allowed simultaneous measurements across an expanded size range, enhancing the characterization of the particle size distribution. Instead of using the standard Palas Sigma-2 passive collector, we used a custom-made directional inlet to align the inlet flow with the mean wind. The exact dimensions of the inlet are provided in Fig. G1. After the inlet, the particles are guided through a sampling tube with drying section IADS

(Intelligent Aerosol Drying System), avoiding condensation effects. We placed the combined instruments at 2.1 m (referred to





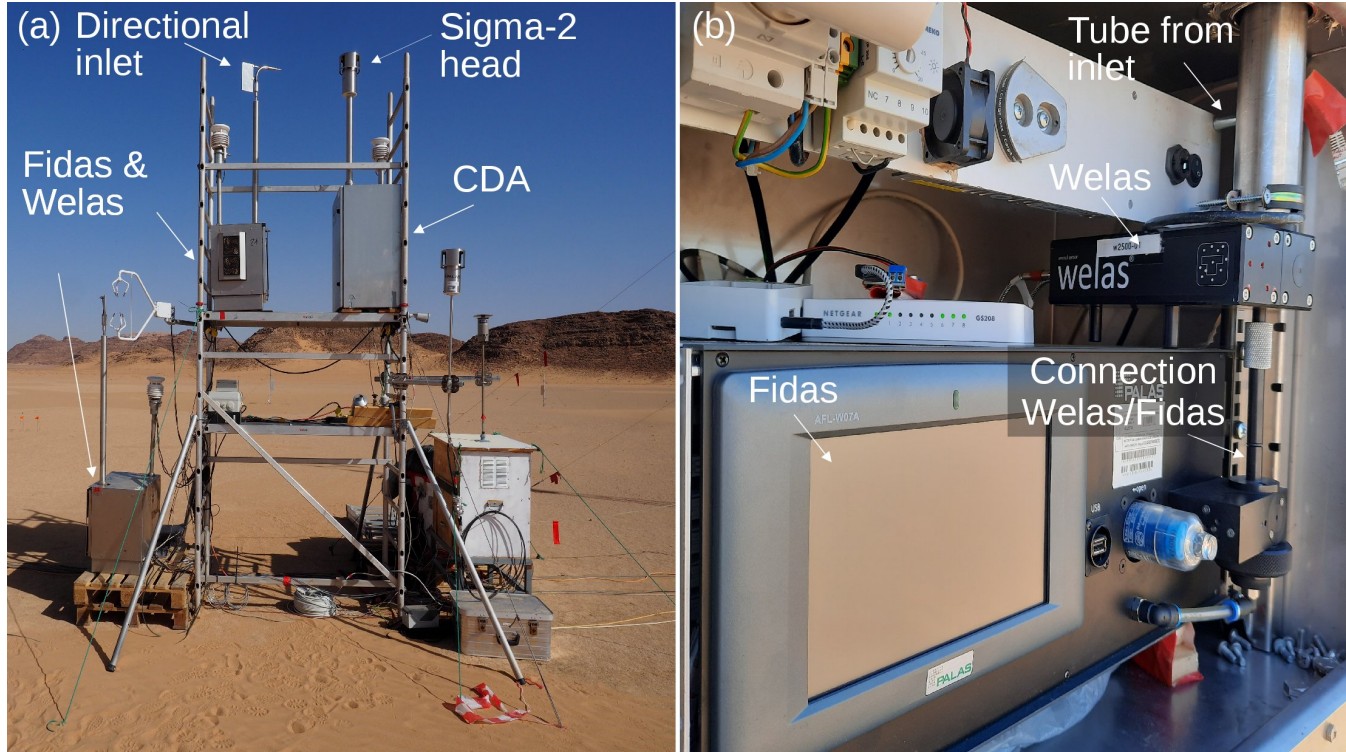

**Figure 6.** Welas, Fidas, and CDA (a) on and next to the scaffolding, (b) Welas and Fidas sharing the same air flow in the metallic box.

as Welas_2m/Fidas_2m) and 3.8 m height (Welas_4m/Fidas_4m) on a scaffolding at a distance of about ~5 m from the rotating mast (Fig. 6a). We recorded single-particle data for the Welas in the 256 raw size bins, i.e. the instrument's raw data before compilation into size distributions. This approach provides high-resolution data, from which we then calculated the PSD. Here, we aggregated the raw 256 bins into 31 approximately logarithmically spaced bins to reduce noise and enhance the clarity of

trends in the PSD. A reduction to 31 bins was sufficient to achieve this in case of the Welas. The Fidas data were analyzed with the PDAnalyze software from Palas GmbH and we summarized the 256 bins into 16 logarithmically-spaced bins similar to the approach in González-Flórez et al. (2023).

**CDA:** Next to Fidas_4m and Welas_4m, a CDA was placed at 4 m height on the scaffolding and set to measure in the size range of $0.8 - 100\,\mu m$ with a flow rate of $5\,l\,min^{-1}$. The CDA uses the same measurement principle as the Fidas and Welas, but

unlike the other two instruments, it senses the entire flow volume rather than just a portion of it. This approach was expected to be beneficial for larger particles, as their number concentrations in ambient air are typically much lower than those of smaller particles. By increasing the sensed volume, and thereby the number of detected particles for any given concentration, this setup should improve the statistical robustness at which the larger, less abundant particles are counted. For the CDA, we also recorded single-particle data (as for the Welas) and rebinned the 256 bins to the same 31 bins as for the Welas. The last bin of the Fidas,



Welas, and CDA were removed because they correspond to the respective upper boundaries of the instruments' measurement range, where limitations in size classification accuracy and potential edge effects introduce uncertainties in the recorded data.

**General data treatment:** The measurements of all aerosol spectrometers were time-synchronized by connecting them to a time server. The devices used different sampling frequencies, mostly 1 Hz, but for the analysis we averaged them over 15 min (time stamps correspond to the end of the interval) in consistency with earlier campaigns (González-Flórez et al., 2023; Panta et al., 2023; Yus-Díez et al., 2023). Note that Dupont et al. (2024) used time stamps that corresponded to the middle of each interval, but that data used here (e.g. $u_*$) was re-assigned to match the time stamps used in this study. A time interval of 15 min was chosen as it is still small enough to identify variations, but also large enough to characterize the boundary layer turbulence spectrum. To derive mass concentration from number concentration, we assumed a particle density of $\rho = 2650\,\mathrm{kg\,m^{-3}}$ as reported by Tegen and Fung (1994).

To gain information about the mineral dust emission, dust fluxes are often used to infer the emitted dust PSD (Shao, 2008). This emission PSD at height zero has never been measured directly. The dust fluxes are typically estimated using the flux-gradient (FG) and eddy covariance (EC) methods, but their applicability for particles $d_p > 10\,\mu\mathrm{m}$ is limited (Fratini et al., 2007; Shao, 2008; Dupont et al., 2021). Recent experiments confirmed that dry deposition can strongly influence both concentration and diffusive flux PSDs, modulated by wind-dependent fetch length and friction velocity (González-Flórez et al., 2023), supporting earlier modeling studies (Dupont et al., 2015; Fernandes et al., 2019). Therefore, in this study, as a first step we approximate the emission dust PSD by the PSD close to emission instead of emission dust fluxes assuming that they will be finer than the emission flux PSD.

To determine a representative central value for each size bin, we used the geometric mean of the upper and lower bin boundaries. Fidas, Welas, and CDA were calibrated using monodisperse polystyrene latex spheres (PSL, MonoDust 1500, manufactured by Palas GmbH). The Fidas and Welas instruments were calibrated at the start of the campaign, while the CDA and UCASS were calibrated prior to shipping (in the case of the UCASS: Girdwood et al., 2025). As a result, the optical diameters used to describe the original instrument PSD correspond to the diameters of latex spheres that produce the same intensity of scattered light as the dust particles being measured. We converted optical diameters into geometric (volume-equivalent) diameters for Fidas and Welas by following the approach proposed by Huang et al. (2021) that was implemented in González-Flórez et al. (2023) for the same Fidas instrument. We assumed an average complex refractive index (CRI) for the Middle East (1.53 - 0.0011i) from Di Biagio et al. (2019), and prolate biaxial ellipsoids shape for the dust particles with an aspect ratio of 1.49. Here, we made use of the single particle scattering calculations for biaxial spheroids from the Gasteiger and Wiegner (2018) database in this diameter conversion procedure. Figure E1 of the Appendix E compares the obtained geometric diameters with the default optical diameters. It reveals that the optical diameters underestimate the dust particle sizes, primarily due to the combined influence of dust asphericity and refractive index. Especially for larger particles the underestimation is substantial, whereas the differences decreases for smaller particles. This conversion to geometric diameter enables easier comparison of our observed with theoretical and modeled PSDs. For the SANTRI2, we converted projected-area into geometric diameters as explained in detail in Appendix E.



**Outlier correction and harmonization**: Throughout our research campaign, we observed irregularities in the mineral dust
PSD across instruments, which presented two main challenges. Some instruments (UCASS, SANTRI2, occasionally Welas,
and CDA) displayed sharp peaks in their size distribution data that were unrealistic and inconsistent with visual observations
(e.g., periods of low wind) and did not align with data from other instruments (or other sensors, in the case of the SANTRI2).
We attribute these anomalies to outliers, likely caused by unwanted light reflections or contaminated sensors.

Outliers are expected due to various instrumental factors: in SANTRI2, light reflected from surfaces can lead to erroneous
detections; in UCASS, a faulty connection could result in recurring high numbers in the counts over time steps without similar
values in the surrounding; and in Fidas, Welas, and CDA, outliers can arise from misclassification errors, where particle counts
are incorrectly assigned to adjacent bins. These discrepancies in PSD underscored the necessity of correcting such outliers to
ensure the accuracy and reliability of subsequent statistical analyses. This step was crucial for uncovering trends and patterns
in the data. To address outliers and harmonize measurements between instruments, various correction methods were applied,
as detailed in Appendices C and D, and summarized below. The outliers detected in SANTRI2 data were managed by applying
a filter based on comparison between counts registered at the different sensors and outlier statistics (Appendix C1). For the
UCASS, we used the count distribution as the basis to remove outliers (Appendix C2). For the Welas and CDA instruments,
we excluded counts recorded in one of the 31 bins (from the summarized 256 raw bins) when no counts were detected in the
preceding smaller bin over a 15-minute interval (Appendix C3).

To establish a baseline for identifying and quantifying systematic differences between the instruments, we conducted inter-
comparison measurements at the end of the campaign. During this phase, all devices, except the SANTRI2, were placed in
close proximity and aligned at ~2 m height. The SANTRI2 devices were installed nearby and next to each other in the setup
for which they were originally designed, i.e. vertically to measure saltating particles. This substantially increased the num-
ber of particles they registered and enabled a more robust statistical comparison. The Welas instruments utilize a light source
with a guaranteed lifespan of 400 hours. Beyond this period, the light spectrum, and therefore also the bin classification, may
shift, which we suspect occurred for Welas_2m. To reconcile discrepancies between the measurements from Welas_2m and
Welas_4m, the bin distribution of the Welas_2m data was stretched based on measurement differences obtained during the
intercomparison period. Further details on this adjustment are provided in Appendix D3. For instruments of which we used
more than one device, such as Fidas, Welas, and SANTRI2, we applied bin-wise linear regression corrections of the number
counts to eliminate systematic biases between devices of the same type. This approach follows the methodology outlined in
González-Flórez et al. (2023) and Dupont et al. (2024), and is detailed further in Appendix D1. To harmonize the PSDs from
instruments with different measurement principles and correct for systematic differences between them, we applied a constant
scaling factor, using Fidas_4m as the reference. The scaling factor was retrieved by minimizing the difference between the
PSDs of Fidas_4m and each of the other instruments in the size range $1 - 10\,\mu m$ during the intercomparison period. The re-
trieved correction factor was then applied to the entire PSD for the corresponding instrument over its measurement period. This
procedure is further detailed in Appendix D2.

Due to high inconsistencies in two SANTRI2 units (high noise level), the UCASS (oscillation) and CDA (decrease for
$d_o > 20\,\mu m$) data, we excluded them from the analysis (for more detail see Sect. 3.2). To create a unified PSD covering the





entire size range, data from the two SANTRI2 (with 5 sensors each), the two Welas and the two Fidas instruments (2 and 4 m height) were combined by aligning overlapping size bins and correcting for inter-instrument differences. To harmonize the PSD across instruments, we applied a rebinning method that interpolates measurements from the original bin edges to a common set of target bins, using bin-weighted averages based on overlapping size ranges. As instruments are less sensitive and reliable toward the edges of their measurement size ranges, we did not use the full size ranges of the individual instruments to combine them into the averaged PSD, and bins without valid contributions were assigned NaN values. The whole procedure is described in detail in Appendix F. For some analysis steps (e.g., comparison to other studies or the comparison of different time steps), to make the averaged PSDs' shape from different time steps comparable, they are normalized over a certain diameter range (Sect. 2.1.2) so that the integral is equal to 1 in every time step. The rebinning method described above was also applied to compare our J-WADI data with results from other field campaigns and was normalized according to the approach outlined by Formenti and Di Biagio (2024).

### 2.1.3 Particle sampling and analysis

Dry deposition samplers of a 'flat-plate' type (FPS; Waza et al., 2019; Panta et al., 2023) were used to collect deposited particles directly on pure carbon adhesive substrates (SpectroTabs, Plano GmbH, Wetzlar, Germany). Briefly, the FPS consists of two circular brass plates, a top plate with a diameter of 203 mm and a bottom plate with a diameter of 127 mm with a distance of 16 mm (Fig. 7). Between the plates the wind is channeled and thus turbulence is reduced. A 25 mm aluminum stub is placed in the center of the lower plate with the adhesive surface level with the plate. As a function of wind speed, particles larger than a few hundreds of µm are generally prevented from reaching the sampling surface due to their large settling velocities (Ott and Peters, 2008). The sampler was placed approximately 10 m from the rotating mast and 1.5 m above ground on a tripod. The substrates were typically exposed for 24 hours with some exceptions during high dust loadings (Table B1).

**Scanning electron microscopy (SEM)** A FEI Quanta 400 FEG ESEM (FEI, Eindhoven, The Netherlands) was used to analyze the morphology, size, and surface features of individual particles by directing a focused beam of electrons onto a sample and detecting the resulting secondary and backscatter electrons (Scanning Electron Microscopy, SEM). The system is operated in a semi-automated way, in which backscatter images are used to detect the particle on the carbon substrate by their higher brightness. For each identified particle, size, shape and an X-ray fluorescence spectrum (using an X-Max 150 energy-dispersive X-ray detector (EDX), Oxford, Oxfordshire, UK) is recorded. The samples were analyzed under high-vacuum conditions without pretreatment. An acceleration voltage of 12.5 kV, a beam current of 18 nA, and a working distance of approximately 10 mm were used. Analysis was carried out at two different magnifications (1.28 and 0.16 µm pixel$^{-1}$). This allowed for the sizing of particles with a minimum projected-area diameter of 0.2 µm at the high magnification. The low magnification enabled the analysis of a large sample area, yielding sufficient counting statistics for larger particles. At higher magnification, analysis locations on the substrates were randomly selected, while at lower magnification the total substrate could be investigated. After the automated analysis, the images were manually inspected for obvious surface defects and the corresponding regions removed from the data (less than 1 % of the surface). Plant fibers found on some samples sized several 100 µm were also disregarded. Also, particles with very low EDX count rates due to shading effects were not included. On average, 3500 particles were



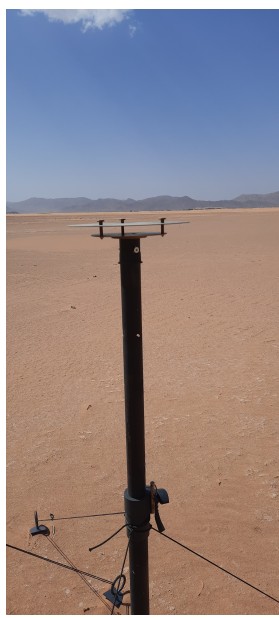

**Figure 7.** Flat-plate sampler (FPS) used in the J-WADI campaign, adapted from Ott and Peters (2008). Positioned $10\,\text{m}$ from the rotating mast, it consists of two brass plates at a distance of $16\,\text{mm}$ and a circular substrate (diameter $= 25\,\text{mm}$) in the center of the bottom plate.

analyzed for each sample (min. 1700). Mass deposition rates were calculated from the observed particles and the analysis area employing different geometric corrections. For details on the procedure refer to Kandler et al. (2018) and Panta et al. (2023).

Detection limits were calculated using $2\sigma$ of the peak intensity, and a final sorting step was applied to remove particles with low X-ray counts due to shading effects. Geometric (volume-equivalent) particles sizes for all diameters were estimated from the apparent projected area and the particle shape as explained in Appendix E.

**Concentration conversion and deposition velocities:** Estimating atmospheric airborne concentrations from the NDR requires certain assumptions regarding the deposition velocity. Sedimentation plays a key role in determining the atmospheric lifetime

of dust particles. The terminal fall velocity of particles describes the rate at which particles settle onto surfaces and reflects the balance between gravitational and drag forces. Numerous expressions exist to estimate the velocity of particle deposition (e.g. Stokes approximation). However, most formulas are poorly suited for super-coarse and giant mineral dust particles ($d_p > 10\,\mu\text{m}$) which have a complex aerodynamic behavior, that deviates from Stokes approximation and the idealized spherical shape (Adebiyi et al., 2023). This limitation is compounded by a lack of experimental data to validate their deposition,

particularly under natural environmental conditions. Adebiyi et al. (2023) compared several expressions to estimate particle settling velocity and concluded that for $d_p = 450\,\mu\text{m}$, which represents the approximate size of the largest particles observed after long-range atmospheric transport (Betzer et al., 1988; van der Does et al., 2018), the measured deposition velocity ranges from approximately 1.5 to $3.5\,\text{ms}^{-1}$. This range aligns with the empirical relationships proposed by, e.g., Cheng (1997) and other expressions used in Adebiyi et al. (2023). The assumptions underlying the estimates of deposition velocity presented

in Adebiyi et al. (2023) may not fully apply to our field conditions. While terminal fall velocity describes the rate at which





particles settle under gravitational and drag forces, the actual settling onto the FPS may be influenced by turbulent diffusion outside or inside the sampler which does not have such a strong dependence on particle size (Guha, 2008) and small lifting events before deposition, through, e.g., flow dynamics around the FPS (e.g., the top plate) and other instruments. Therefore, the particles might not reach the terminal fall velocity. Consequently, formulas for particle settling velocity – including those
reviewed by Adebiyi et al. (2023) – might not accurately represent the deposition behavior of the particles. Waza et al. (2019) and Panta et al. (2023) reported that with none of the traditional deposition velocity expressions the shape of aerosol spectrometer and FPS could be matched, but instead the deposition velocities appeared to have a much lower particle size dependency, i.e., their general shape were similar. In our study, we could confirm that none of the formulas could match the shapes of FPS with aerosol spectrometer and therefore also assumed a uniform deposition velocity for all particle sizes. A constant deposition
velocity of $v_d = 0.0007\,\mathrm{m\,s^{-1}}$ provided the best fit to our data as found by minimizing the sum of squared differences.

## 3    Results

This section provides an overview of meteorological conditions and measured dust mass concentrations during the campaign (Sect. 3.1). We present mass concentration PSDs from active dust emission events before and after applying correction methods (Sects. 3.2–3.3) and discuss discrepancies between instruments, including inlet efficiency estimates. The aerosol spectrometer
PSDs are compared with FPS and SEM results (Sect. 3.6), followed by an analysis of PSD variability in relation to friction velocity and atmospheric stability (Sect. 3.5). Finally, we compare our findings with other field measurements (Sect. 3.3).

### 3.1    Meteorological and dust conditions during the campaign

The meteorological conditions during the campaign (Fig. 8) are characterized by high air temperatures, with mean daily minimum and maximum averages of $\overline{\theta}_{\mathrm{daily\ min}} = 18.2\,^{\circ}\mathrm{C}$ and $\overline{\theta}_{\mathrm{daily\ max}} = 32.4\,^{\circ}\mathrm{C}$, respectively (Fig. 8a). Relative humidity at 4 m
height was continuously low, with average daily minimum and maximum averages of $\overline{\mathrm{RH}}_{\mathrm{daily\ min}} = 20\%$ and $\overline{\mathrm{RH}}_{\mathrm{daily\ max}} = 61\%$ (Fig. 8b). No precipitation was recorded during the campaign. The mean atmospheric pressure during the campaign was 924 hPa, exhibiting a diurnal cycle with the lowest values occurring around 0 and 12 UTC (local time = UTC+03:00 h local time), and two maxima, one around 6 and the other around 21 UTC (Fig. 8c).

Local dust emissions predominantly occurred between 11 and 16 UTC at wind speeds of more than about $v_{4\,m} \approx 6\,\mathrm{m s^{-1}}$,
with usually north-westerly wind directions (Fig. 8 d, e, h). The 15 minute averaged dust mass concentration time series is shown in Fig. 8h. It represents the average of the two SANTRI2s (with 5 sensors on each), the two Welas and the two Fidas instruments (2 and 4 m height), including the correction procedure explained in Sect. 2.1.2 and in further detail in Appendices C-F. Results obtained with each instrument and more detail on the combined PSDs are presented in Sect. 3.4. Several events with high total mass concentrations ($C_m > 1 \times 10^4\,\mathrm{\mu g\,m^{-3}}$) were recorded, with significant contributions of particles with
diameters greater than 20 μm and also regularly larger than 60 μm. For higher total mass concentrations, typically also more larger particles contributed to the mineral dust mass. The most continuous dust event occurred on 29 September 2022 with intensive dust emission between 10 and 15:30 UTC and high total mass concentrations up to $C_m = 10^5\,\mathrm{\mu g\,m^{-3}}$.



**Figure 8.** 15-minute averaged meteorological and dust conditions during the campaign. Dashed lines (a-e, mast data) and red dots (f-g, scintillometer data) indicate missing values in the measurements used that were filled with data from other instruments as described in Sect. 2.1. (a) Temperature (4 m), (b) relative humidity (2 m), (c) pressure (1 m), (d) wind speed (4 m), (e) wind direction (4 m), (f) Friction velocity $u_*$ from the scintillometer (2.54 m). (g) Atmospheric stability represented by $z/L$, where $L$ is the Obukhov length from the scintillometer and $z$ is the reference height 2.54 m. (h) Particle mass concentration by particle size averaged for Welas, Fidas, and SANTRI2. These data include the corrections explained in Sect. 2.1.2. Color shading represents the mass concentration in each size bin, whereas the red line indicates the total dust concentration summed over all bins.

We obtained a threshold friction velocity of $u_{*t} = 0.22\,\mathrm{m\,s^{-1}}$ using the Time Fraction Equivalence Method (Sect. 2.1.1). We note that saltation was occasionally registered already at lower friction velocities (Appendix A). This discrepancy could be due to intermittent saltation not captured within the 15-min periods used to derive $u_{*t}$, or localized variations in surface conditions.






During periods when the threshold friction velocity was exceeded ($u_* > 0.22\,\mathrm{m\,s^{-1}}$, typically around noon or in the afternoon in UTC time, Fig. 8f, gray line), noticeable contributions from particles with diameters ($d_p$) exceeding 62.5 μm (giant mineral dust particles) were observed. Additionally, several particles > 80 μm in diameter were registered. For unstable and neutral conditions ($z/L \leq 0$), the majority of more intense dust events were observed, particularly during the transition from

unstable to neutral conditions (Fig. 8g). A more detailed analysis of the sensitivity of observed dust mass concentrations to stability and friction velocity is presented in Sect. 3.5.

## 3.2   Uncorrected aerosol spectrometer size distributions

We present the uncorrected 15-minute average PSDs of mass concentration of the two Fidas, Welas, UCASS, and the one from the CDA with optical diameters, and that of the two SANTRI2 with projected-area diameter in Fig. 9a. The diameters to

which the mass concentrations are assigned are the geometric mean of the upper and lower bin boundaries. The PSDs highlight significant mass concentrations of particles larger than 10 μm, with pronounced peaks between approximately 10 μm and 30 μm in diameter. The 15-minute average PSDs from each instrument demonstrate substantial temporal variability, as indicated by the error bars, which represent the standard errors within the corresponding averaging period, and for SANTRI2s also across the analyzed sensors per instrument.

PSDs from Fidas_4m and Welas_4m are consistent for particle diameters smaller than 7 μm (Fig. 9a). Contrary to our expectations of higher dust concentrations and larger particles closer to the ground, Welas_2m PSDs show peak mass concentration at a smaller diameter of approximately 20 μm compared to Welas_4m, which peak at around 30 μm (Fig. 9a). For particles smaller than 10 μm, Welas_4m and Welas_2m alternate in exhibiting higher mass concentrations. In contrast to the Welas instruments, the behavior of the two Fidas PSDs aligns with expectations, as Fidas_4m generally shows lower concentrations

across most of the size range. However, toward the upper end of the size range, the concentration of Fidas_4m partially exceeds that of Fidas_2m (e.g., Fig. 9a 2022-09-29 14:45). Another characteristic of the Fidas mass concentration PSDs is the absence of a distinct peak. Instead, their mass concentration appears to be relatively evenly distributed across sizes from approximately 10 μm in Fig. 9a. Mostly, UCASS B and Welas_4m generally show good agreement. The measurements of UCASS B closely match those of Welas_4m for particle diameters $d_p < 4$ μm and up to $d_p \approx 10$ μm. However, the PSD measured by UCASS B

exhibits oscillations instead of forming a smooth curve up to approximately $d_p = 10$ μm. Beyond this range, UCASS B shows a more gradual increase in concentration compared to Welas_4m, although both instruments maintain similar distribution shapes. Both instruments exhibit peak concentrations in the particle size range of $d_p = 20$ to 30 μm, with the peak concentration of UCASS B being roughly an order of magnitude smaller than that of Welas_4m. UCASS A exhibits significant differences in its mass concentration PSD compared to the other instruments. At approximately 1 μm, its mass concentration is comparable

to that of the other instruments, but it increases by an order of magnitude for the third bin (around 1.3 μm). For larger particles, the concentration decreases until it matches the concentrations of the other instruments at ~2 μm and increases again at ~3 μm to an order of magnitude higher than the concentrations of other instruments (at ~4 μm) and oscillates around the other instruments' PSDs for $d_p$ up to ~10 μm. At 10 μm, UCASS A's mass concentration aligns well with the UCASS B concentrations. The oscillations of the PSD, i.e., the classification of size bins, remains an open question, as further adjustments may





**Figure 9.** 15 minute average size distributions of mass concentration for dusty conditions on 29 September 2022 for 3 subsequent 15-min time periods from 14:30 until 15 UTC. (a) Uncorrected PSD with optical diameters (except SANTRI2, which uses projected-area diameter) and (b) corrected PSDs with geometric diameters. Standard errors are indicated by vertical lines (only positive errors are shown). Average 4 m friction velocity $u_*$ for each 15-min period are indicated in the panel titles. Dashed lines indicate the size ranges of the dust size classifications (Adebiyi et al., 2023).

be needed to optimize particle categorization and is therefore excluded from further analysis. The CDA PSDs (measured at 4 m height) show significant differences compared to the PSDs of the other instruments. Before peaking at around 15 μm, the CDA concentrations generally agree well with those of Fidas_4m and Welas. After the peak, the CDA concentration decreases rapidly, crossing below Fidas' mass concentration and eventually falling below all other measurements. This decrease could not be resolved by any correction method. We suspect that this decrease was due to either a reduced sampling efficiency of





the Sigma-2 inlet in that size range or a lower sensitivity for larger particles, which was not the case for the other instruments.
Consequently, the CDA data were also excluded from further analysis.

In certain cases, the SANTRI2 PSDs align well with the extended particle size trends observed with the other instruments
(Figs. 9a 2022-09-29 14:45, 15:00). However, in Fig. 9a 2022-09-29 14:30, the two SANTRI2 units, each averaged over the
five sensors, show higher mass concentration PSDs than expected, compared to measurements from the other instruments.

It is important to note that the SANTRI2s recorded projected-area diameter whereas the other instruments recorded optical
diameters which could potentially change the agreement between SANTRI2 and other instruments' PSDs.

## 3.3    Corrected size distributions

The uncorrected PSDs with optical diameter shown in Fig. 9a reveal the original measurements taken by our instruments,
indicating potential biases and inaccuracies due to low sampling efficiencies and variability between instruments. The data

presented in this Section was corrected as explained in Appendix C-E. Here, we discuss the corrected PSD mass concentrations
as shown in 9b and the remaining variability between instruments. Overall, the comparison between Figs. 9a and 9b highlights
that the correction procedures result in more consistent concentrations across instruments, although we could not eliminate all
sources of discrepancy.

After correction, Welas_2m consistently show higher values than Welas_4m, with some exceptions for particles larger than

60 μm (Fig. 9b 2022-09-29 14:45, 15:00). Both instruments exhibit a peak at approximately the same diameter (~50 μm). After
correction, they better match the concentrations observed in the Fidas measurements, particularly for particles larger than $d_p =$
1.2 μm and up to ~10 μm. In the Fidas' mass concentrations, a plateau in measurements for particles larger than $d_p \approx 12$ μm is
visible, which we attribute to potential limitations in measuring larger particles – limitations that could not be corrected by any
of the correction mechanisms applied. In comparison to the uncorrected PSDs of the SANTRI2s, most of the corrected PSDs

now better fit the prolongation of the other instruments. In Figs. 9b 2022-09-29 15:00, the two SANTRI2 present lower mass
concentration PSDs than would be expected from the Welas but fit well the overall appearance. For further analysis, only the
SANTRI2, Welas, and Fidas instruments were considered, as most of the differences between these instruments were resolved.
They were also used for the combined overall PSDs shown in the next subsection.

## 3.4    Possible reasons for discrepancies between aerosol spectrometers

The observed differences in the uncorrected and corrected PSDs presented in Sects. 3.3 and 3.2 can be attributed to several
instrumental factors.
**The use of inlets:** The use of inlets for aerosol sampling significantly influences the measurements. All instruments equipped
with inlets, such as the Welas, Fidas, and CDA, experience sampling inefficiencies, particularly for larger particles, due to
losses within the inlet system (Kulkarni et al., 2011). In contrast, open-path instruments without inlets, do not suffer from inlet

losses, but may be more susceptible to environmental interference. The sampling efficiency, $\eta$, is influenced by the inlet or
pipe design, flow dynamics, and particle characteristics. The inlet losses of the different instruments used here are described
in Appendix G. The directional inlet of Fidas and Welas was characterized using empirical models explained in detail in Ap-



pendix G3. Their inlet efficiency for different wind conditions is shown in Fig. 10a. For wind speeds $v \leq 5$ m s$^{-1}$ and particle diameters $d_p \leq 5$ μm, the efficiency $\eta$ is approximately 100%, decreasing to 0% at $d_p \approx 30$ μm. For wind speeds $v > 5$ m s$^{-1}$ and particle diameters $d_p > 5$ μm, the efficiency $\eta$ increases, peaking at $d_p \approx 12$ μm, and then decreases to 0% at $d_p \approx 40$ μm. The peak for wind speeds $v = 11$ m s$^{-1}$ is even at sampling efficiencies of 140%, so an oversampling of particles $d_p \approx 12$ μm occurs. Most of the losses stem from gravitational settling in the horizontal part of the pipe or impacts due to the bend. A similar inlet design with comparable dimensions was previously quantified by Schöberl et al. (2024). They reported cut-off diameters (defined as 50% loss) smaller than 10 μm. In contrast, our calculations show a cut-off diameter of approximately 30 μm for $v = 5$ m s$^{-1}$. The lower cut-off diameters observed by Schöberl et al. (2024) may be attributed to their slightly different inlet dimensions (inner diameter = 4.527 mm), the calculation of the bend efficiency $\eta_{bend}$ in degrees instead of radians (as noted in the supplementary material of Schöberl et al. 2024), or the high flow velocities and Reynold numbers associated with their airplane measurements. By dividing the mass concentration PSDs of the Fidas and Welas instruments by the sampling efficiencies $\eta$ of the directional inlet under the measurement conditions, corrected PSDs can be estimated, as shown for an example PSD in Fig. 10b. For the Welas, no significant change in concentrations is observed for $d_p < 20$ μm, not even for the oversampling which occurs at $d_p \approx 12$ μm. However, for larger particle sizes, the corrected PSDs are clearly increased by several orders of magnitude compared to the uncorrected ones. A similar trend is observed for the Fidas, although most of the Fidas size range remains unaffected by large inlet inefficiencies as their size range stops before $d_p = 50$ μm. The estimated inlet efficiencies suggest that almost no particles larger than around 20 μm should have been detected, yet our results show the measurement of a significant number of particles in this size range. The empirically estimated inlet efficiencies therefore appear unrealistic. The underestimation of inlet efficiencies for large particles could potentially result from neglecting the re-emission of particles that initially settled, a process not accounted for in the applied formulas. Additionally, traditional deposition schemes may overestimate gravitational settling for large particles (Adebiyi et al., 2023), highlighting potential limitations in the modeled particle dynamics. Furthermore, the underestimation may also stem from limitations in the applicability of the used formulas, which might not be entirely suitable for our context – for instance, due to the presence of particles that are so large that the Stokes number regimes, for which the expressions are valid, is exceeded. Results from application of the formulas beyond their valid range are indicated by dashed lines in Fig. 10a, overlapping with diameter ranges that have low $\eta$.

The UCASS is a passive instrument and in principle open-path (i.e. inlet-free), however its cylindrical shape may act similar to an inlet. Limited information about its sampling efficiency $\eta$ is available beyond the findings of Girdwood et al. (2022), who reported low losses for droplet diameters between 3 and 10 μm. Flow dynamics simulations by Smith et al. (2019) indicated that the air velocity in the sampling area is approximately 12% higher than the ambient air velocity for an ambient wind speed of 5 m s$^{-1}$. Their results showed no significant turbulence inside the instrument and good sampling efficiency for particles smaller than 40 μm. Therefore, due to its large opening (5 cm on the smaller side), significant losses in the nozzle are not expected. When applying the formulas described in Appendix G to a simplified geometry of the UCASS (*i.e.,* assuming a round instead of a oval opening, and no electronics inside the tube to disturb the flow), we found gravitational efficiencies ($\eta_{grav}$) close to one, but substantial losses due to turbulent inertial deposition ($\eta_{\text{turb-inert}}$). This phenomenon occurs when large particles, owing to their high inertia, are unable to follow the curved streamlines of turbulent eddies and are deposited on the





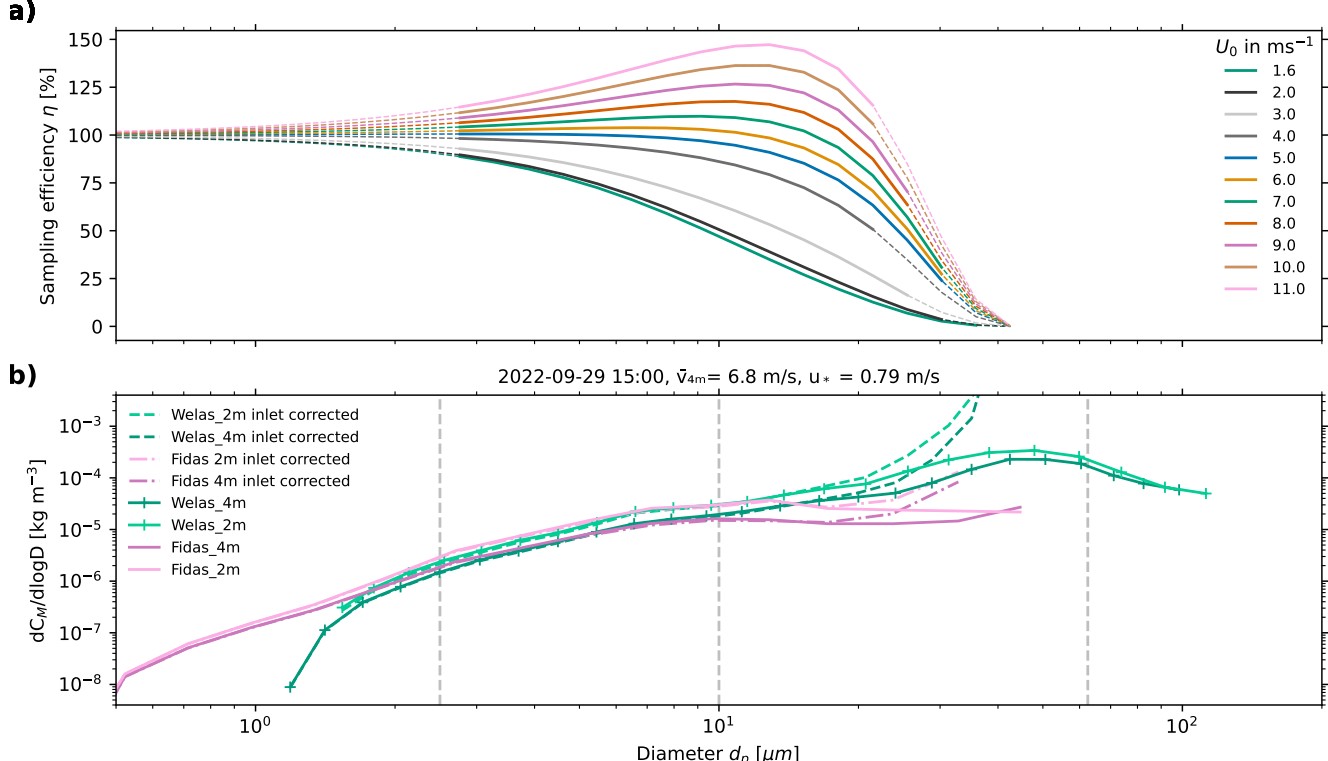

**Figure 10.** Sampling efficiency $\eta_{sampling}$ (a) for different wind conditions $U_0$ for the directional inlet of Welas and Fidas. Dashed curves indicate that the applied formulas may not be valid for the respective diameter range and wind conditions. (b) Example PSD (solid lines) together with the corrected PSD corrected by the sampling efficiency $\eta_{sampling}$ for Welas and Fidas in dashed lines. Vertical dashed lines indicate the size ranges of the dust size classifications (Adebiyi et al., 2023).

walls of the instrument. For higher wind speeds around $10\,\mathrm{m\,s^{-1}}$, inlet efficiencies rapidly decrease from approximately 90% for $11\,\mu\mathrm{m}$ particles to nearly 0% for particles of $20\,\mu\mathrm{m}$ due to the large pipe diameters, high flow rates, and resulting high

495 Reynolds numbers (Eq. G9). These findings highlight a discrepancy in the turbulent flow in the UCASS between the empirical formulas and simulations, particularly for larger particles, which may stem from simplified assumptions in the modeled particle dynamics and the omission of re-suspension effects in the formulas, as partly discussed in Kulkarni et al. (2011).

  Given the limitations of the calculation for the directional inlet and the UCASS housing and the apparent mismatch between the theoretical/empirical estimates of inlet efficiencies and the observed particle counts, the inlet efficiencies will not be applied

500 for the correction of the PSD in the following analysis to avoid introducing additional uncertainties, particularly for particles larger than $10\,\mu\mathrm{m}$. Instead, the results will be interpreted with awareness of potential losses due to turbulent inertial deposition (UCASS) and gravitational settling in sampling pipes or bends (directional inlet). The inlet efficiencies of the instruments will be investigated in more detail in the future using numerical modeling of the flow dynamics in the inlets.



Operating without any inlet, the SANTRI2 relies on the wind field to guide the particles through the optical path. Although this design avoids inlet-induced biases, turbulence effects caused by the (quite slim) platform to which the sensors are attached are possible. Despite this, the approach offers the most direct and unaltered sampling of ambient aerosol among our instruments, providing insights into the nearly undisturbed characteristics of large dust particles.

**Measurement principle differences:** A second reason for the discrepancies in PSDs between instruments lies in their measurement principles, such as optical scattering (used by Welas, Fidas, CDA, and UCASS) versus the optical gate mechanism employed by SANTRI2, which introduces additional variability as it measures the projected area. Optical instruments estimate particle size based on light scattering, which can be influenced by factors such as particle shape, composition, and refractive index. The various devices based on optical scattering differ in aspects like scattering angle, sensor area, and light source, which can lead to inaccuracies, especially for non-spherical or irregularly shaped particles. In contrast, optical gate devices determine particle size by measuring the shadow cast on a photodiode, meaning the obtained 2D shape for non-spherical particles is highly dependent on their orientation when illuminated. In order to overcome these limitations, we harmonized measurements from the different devices and transformed the particle sizes to geometric diameters, assuming biaxial ellipsoids. However, some of the aforementioned causes of uncertainties, such as particle shape and refractive index, remain unresolved. In Appendix H, the results for assuming triaxial instead of biaxial ellipsoids are shown.

**Additional differences:** The classification of particle size bins make use of different theoretical frameworks. For optical diameter measurements, Mie theory assuming spherical particles is commonly applied (e.g., Welas, Fidas, and CDA), whereas for the retrieval of projected area diameters, the projected area on the instrument is used. For the transformation to geometric diameters ellipsoidal particles are assumed, which are either based on databases for different CRI (e.g., Gasteiger and Wiegner, 2018, i.e., sensitive to assumed CRI) or on geometric calculations as described in Appendix E. These different approaches influence the shape of the retrieved PSDs.

Moreover, the instruments differ in how they handle partially illuminated particles: the Welas, Fidas, and CDA avoid them by their measurement principle, the UCASSs account for this in its calculations but the SANTRI2s do not.

In addition, the size ranges covered by different instruments introduce variability in accuracy towards the edges of these ranges. Instruments optimized for detecting fine particles may exhibit reduced accuracy and sensitivity for larger particles towards the edge of their size range, and vice versa. This discrepancy is particularly evident in the overlap regions where the detection capabilities of different instruments intersect, resulting in inconsistencies in PSDs.

Finally, the location of instruments can affect recorded PSDs due to proximity to emission locations, atmospheric conditions, and particle transport dynamics. Differences in height and positioning can cause variations in sheltering, turbulence, and detected dust concentrations. For Fidas and Welas, these differences should be minimal since they share the same volume and are separated only by a tube. However, discrepancies may arise if particles are trapped in the tube connecting both instruments (Fig. 6b), potentially reducing counts in the Fidas, though tube clogging was not observed during the campaign. After applying our correction steps, Fidas and Welas concentrations agreed well for $d_p < 10\,\mu m$, but discrepancies arose for larger particles, with Welas concentrations being up to an order of magnitude higher at the upper limit of its size range. It is unlikely that particles with $d_p > 10\,\mu m$ continuously got trapped before reaching the sensors of the Fidas, as we conducted measurements





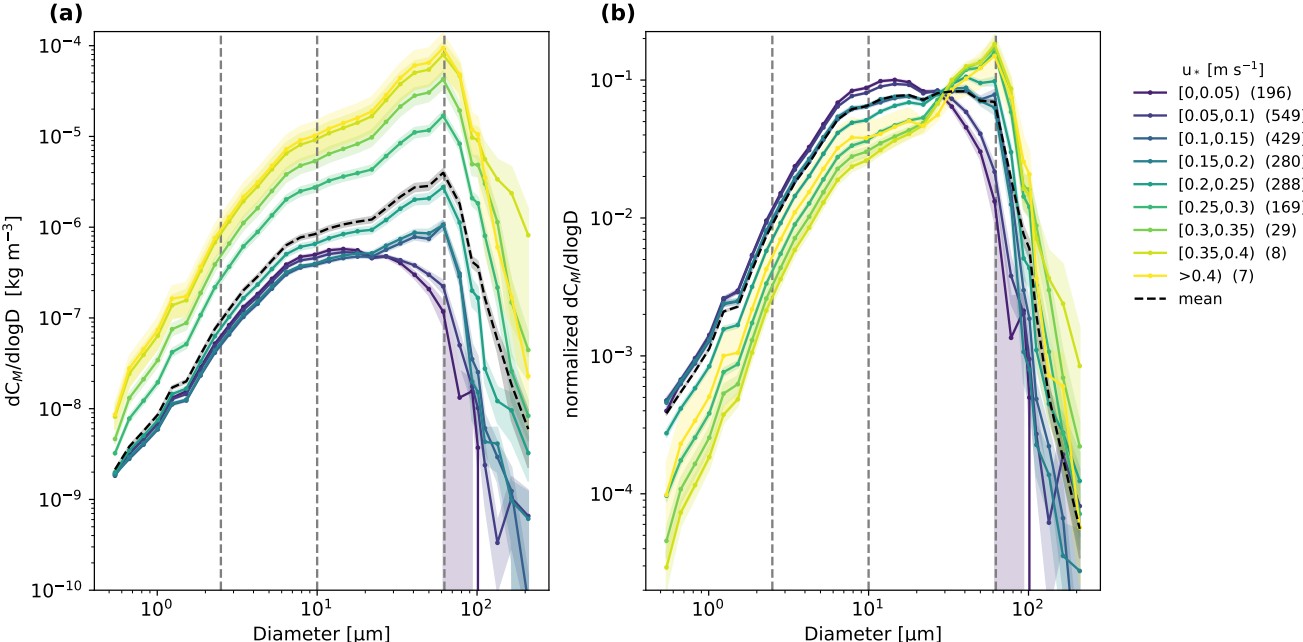

**Figure 11.** (a) Variability of mass concentration PSD with $u_*$ deduced from SANTRI2, Welas and Fidas over the whole campaign time. Colors indicate $u_*$ during the 15-min averaging time period corresponding to the PSDs. Shaded areas depict the standard error of PSDs within each class across different time steps, and the black dashed line the mean of all PSD. Numbers in parentheses indicate the number of 15-min PSDs taken into account in each $u_*$ range. Dashed lines indicate the size ranges of the dust size classifications (Adebiyi et al., 2023). (b) Same as (a) but normalized to unity in each time interval.

over several weeks with the instruments, and no impacts of enhanced blockage were evident from the Fidas measurements.

Therefore, we attribute most of these differences to discrepancies in the instruments' sensitivities (especially at the edges of the instruments size ranges).

Overall, these instrumental differences underscore the importance of employing a suite of complementary measurement techniques to achieve a comprehensive and robust characterization of the full PSD, particularly in the challenging super-coarse and giant particle size ranges. Understanding these differences and their implications is crucial to improving the reliability of

dust measurements and developing better calibration and correction methodologies.

### 3.5 Variability of particle size with $u_*$ and stability

Figure 11a shows the 15-minute averaged PSDs calculated across SANTRI2, Welas and Fidas over the entire measurement period and categorized into different $u_*$ ranges similar to González-Flórez et al. (2023). While the mean of all PSDs in Fig. 11a, indicated by a dashed black line, shows the peak at around $60\,\mu m$, the categorized PSDs differ in their shape and height.

As expected, higher $u_*$ values correlate with an increase in mass concentration ($dC_M$) across all particle diameters (Fig. 11a).




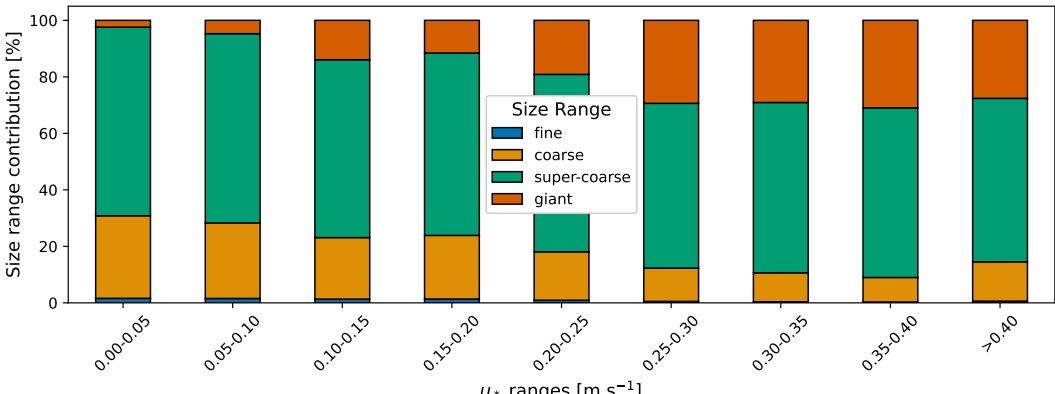

**Figure 12.** Percentage mass concentration abundance of particle size ranges deduced from SANTRI2, Welas and Fidas over the whole campaign time with $u_*$.

However, for lower $u_*$ values ($< 0.2\,\mathrm{m\,s^{-1}}$), the PSDs remain largely consistent. Differences emerge at $d_p = 30\,\mu\mathrm{m}$, where concentrations corresponding to lower $u_*$ values decrease more rapidly – except in the $0.1\,\mathrm{m\,s^{-1}} \le u_* < 0.2\,\mathrm{m\,s^{-1}}$ range where a peak at around $d_p = 60\,\mu\mathrm{m}$ is visible – suggesting that some larger particles were already effectively lifted and detected at friction velocities ($> 0.1\,\mathrm{m\,s^{-1}}$) and below the calculated threshold friction velocity ($u_{*t} = 0.22\,\mathrm{m\,s^{-1}}$). For particles $d_p > 60\,\mu\mathrm{m}$,

the standard error increases significantly, casting doubt on the reliability of this relationship. Starting from $u_* \ge 0.25\,\mathrm{m\,s^{-1}}$, which is above the threshold friction velocity, significantly higher mass concentrations are observed, although the shape of the PSD remains largely consistent as for $0.2\,\mathrm{m\,s^{-1}} \le u_* < 0.25\,\mathrm{m\,s^{-1}}$. For $u_* \ge 0.35\,\mathrm{m\,s^{-1}}$, only a small number of PSD samples is available, but the PSD for the two largest $u_*$ categories are very similar, except for $d_p > 100\,\mu\mathrm{m}$, where concentrations for $u_* > 0.4\,\mathrm{m\,s^{-1}}$ fall behind those observed at lower friction velocities. The SANTRI2 were the only devices operating for

$d_p > 100\,\mu\mathrm{m}$. In this size range, the PSDs show generally lower concentrations at smaller $u_*$ values, but the behavior of $dC_M$ becomes less consistent as $u_*$ decreases, either increasing or decreasing with friction velocity. Especially for $u_* < 0.05\,\mathrm{m\,s^{-1}}$, the concentrations are decreasing to almost zero at $\sim d_p > 100\,\mu\mathrm{m}$. In this friction velocity range, the presence of super-coarse and giant particles is expected to be low. Additionally, these particles may not be captured by the Welas due to inlet inefficiencies. However, the open-path approach of the SANTRI2 allows for direct sampling, increasing the likelihood of detecting these

(few) larger particles which might explain the abrupt change in mass concentration.

Figure 11b shows mass concentrations normalized to unity (15-minute PSDs were first normalized, then averaged). Here, the relative amount of the different particle sizes can be observed and shows more prominently the shift in peak mass concentrations for different $u_*$. The slope of the concentrations are relatively similar up to about $10\,\mu\mathrm{m}$. However, the concentration peak shifts slightly from $12\,\mu\mathrm{m}$ to $60\,\mu\mathrm{m}$ between $0.2\,\mathrm{m\,s^{-1}}$ and $0.3\,\mathrm{m\,s^{-1}}$, which is in line with the threshold friction velocity, whereas

the non-normalized ones already show this shift at $0.1\,\mathrm{m\,s^{-1}} \le u_* < 0.15\,\mathrm{m\,s^{-1}}$. This discrepancy is likely driven by variations in mass concentrations, where higher concentrations have a stronger influence on the averaging process when the PSDs are not normalized.



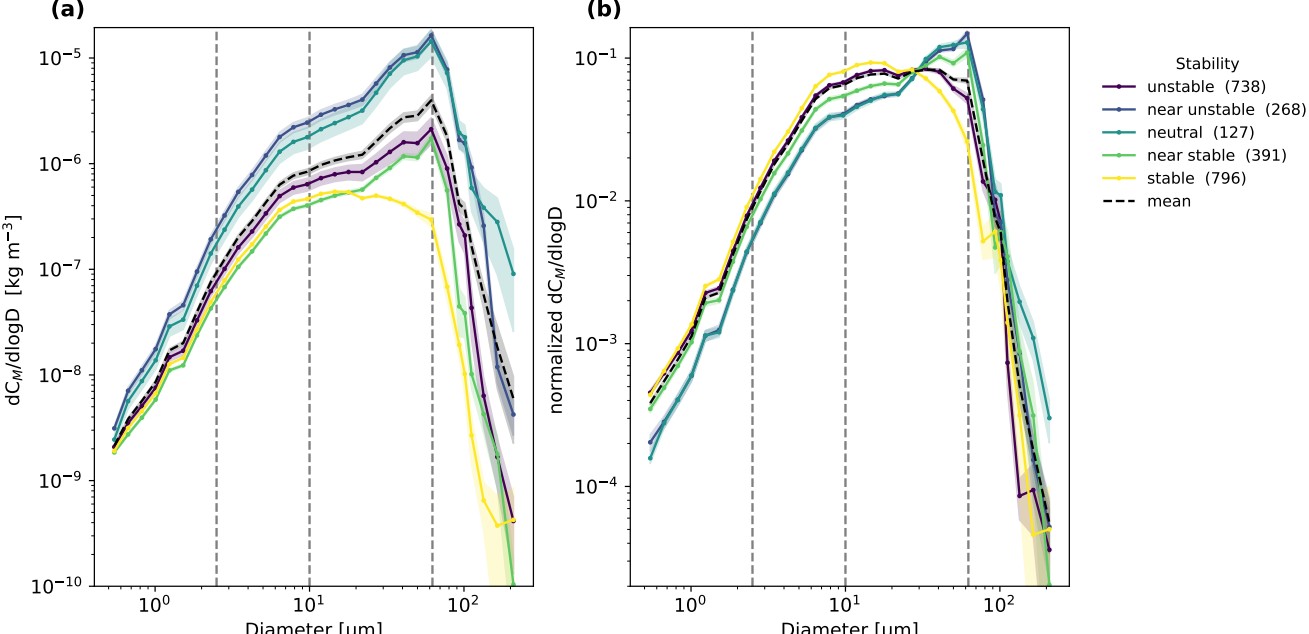

**Figure 13.** (a) Variability of mass concentration PSD deduced from SANTRI2, Welas and Fidas over the whole campaign time with atmospheric stability. The colors indicate different stability ranges and shaded areas the standard error of PSDs within each class across different time steps, and the black line the mean of all PSD. Numbers in parentheses indicate the number of PSDs available within each stability class. Dashed lines indicate the size ranges of the dust size classifications (Adebiyi et al., 2023). (b) Same as (a) but normalized to unity in each time interval.

Variations in the normalized abundance of particles in different size ranges with varying $u_*$ can also be observed in Fig. 12, which shows the total mass concentration contribution of fine, coarse, super-coarse, and giant particles across different $u_*$ cat-
egories. At low friction velocities ($u_* < 0.2\,\mathrm{m\,s^{-1}}$), less than 10% of the mass concentration is contributed by giant particles, and approximately 60% by super-coarse particles. Below the threshold friction velocity ($u_{*t} = 0.22\,\mathrm{m\,s^{-1}}$) recorded dust might be due to intermittent releases or due to dust advected from nearby sources. Dust that occurred at $u_* < 0.22\,\mathrm{m\,s^{-1}}$, however, already contained a great amount of super-coarse and giant particles (see Fig. 12). As $u_*$ increases up to $0.4\,\mathrm{m\,s^{-1}}$, the contribution of giant particles rises to about 20%, while super-coarse particles contribute slightly over 60%. For $u_* > 0.4\,\mathrm{m\,s^{-1}}$, the
contributions of both super-coarse and giant particles decrease slightly, possibly due to low statistics or the presence of potential outliers. This ambiguous dependency suggests that factors other than friction velocity alone may influence the concentration of larger particles in the PSD.

Fig. 13a shows 15-minute averaged PSDs over the entire measurement period, calculated across SANTRI2, Fidas, and Welas and categorized into different stability regimes (Sect. 2.1.1). The total mass concentration across all bins is greatest during near
unstable and neutral conditions, followed by unstable conditions. The lowest mass concentrations are predominantly observed





at night under stable and near stable conditions when friction velocities are small (Figure 8). For stable conditions, very few super-coarse and giant particles are present.

Fig. 13b shows the normalized PSDs. For stable conditions, smaller particles with diameters less than $20\,\mu m$ are most abundant, followed by unstable, and near stable conditions. As stable stratification suppresses turbulence, the lifting and transport
of larger particles is limited, while it allows smaller particles to remain suspended longer. In contrast, for particles larger than $20\,\mu m$ but smaller than $90\,\mu m$, the opposite trend is observed and they are more present for neutral and near (un)stable conditions. No clear trend in stability is apparent for particles larger than $90\,\mu m$. Further investigation is necessary to determine whether this behavior is due to instrument inaccuracies, limited particle statistics or other reasons.

Atmospheric stability and $u_*$ are strongly interconnected as $L$ depends on $u_*$ and for large $u_*$, conditions become in-
creasingly neutral. To investigate the dependency of mass concentration on stability while accounting for the interdependence between $z/L$ and $u_*$, Fig. 14 presents mass concentration as a function of $u_*$ and colored by $z/L$ across different particle size ranges. For $u_* < 0.2\,\mathrm{m\,s^{-1}}$, the majority of the total mass concentrations are below $10^{-6}\,\mathrm{kg\,m^{-3}}$ (Fig. 14a). Across all size ranges, stable conditions correspond to the lowest friction velocity values (mostly $u_* < 0.15\,\mathrm{m\,s^{-1}}$), with the smallest mass concentrations observed. With increasing $u_* > 0.1\mathrm{m\,s^{-1}}$, conditions become more unstable to neutral. Between
$u_* = 0.1\,\mathrm{m\,s^{-1}}$ and $u_* = 0.2\,\mathrm{m\,s^{-1}}$, (near) stable and (near) unstable conditions are present. For larger $u_*$, mass concentrations increased sharply with slight increases in $u_*$ (Fig. 14a). The majority of these data points is categorized as unstable conditions with some near unstable and near stable conditions. These near unstable time periods, however, tended to have slightly higher mass concentrations for a given $u_*$. For instance, for $u_* = 0.25\,\mathrm{m\,s^{-1}}$ and for near unstable conditions, mass concentrations could reach approximately one order of magnitude higher than the average for unstable conditions. However, most of the
near unstable and near stable time period data points gather with the unstable conditions. For $u_* > 0.3\,\mathrm{m\,s^{-1}}$, mostly neutral conditions were registered with a potentially lower mass concentration than would be expected for the elongation of unstable conditions, but with a lack of a clear pattern due to few data points. For all size ranges (Fig. 14b-e), the trends for (near) unstable and (near) stable conditions aligns with that of the total mass concentration (Fig. 14a). However, for giant particles, generally less data points exists. Most data points cluster between $u_* = 0.2\,\mathrm{m\,s^{-1}}$ and $u_* = 0.4\,\mathrm{m\,s^{-1}}$ for unstable conditions.
For neutral conditions, the mass concentration started to increase at higher $u_* > 0.3\,\mathrm{m\,s^{-1}}$ (Fig. 14e).

We observed a slight trend for increased mass concentrations with near unstable followed by unstable and neutral conditions, but no clear pattern emerged. Our results neither fully support nor contradict previous findings by Khalfallah et al. (2020), Shao et al. (2020) or González-Flórez et al. (2023), and Dupont (2022). Further investigation is required to fully understand these dynamics. However, it is important to note that we analyzed dust concentration PSDs rather than fluxes, as done in these
studies.

## 3.6 Comparison between particle size distributions from aerosol spectrometers and FPS

To confirm the PSDs obtained with aerosol spectrometers (Sect. 3.3), we compared our results against those derived from physical samples collected using FPS and analyzed through SEM. This comparison allowed us to evaluate the accuracy of the spectrometers, particularly for larger particle sizes where instrumental biases, such as inlet efficiencies and optical corrections,







**Figure 14.** Mass concentration deduced from SANTRI2, Welas and Fidas over the whole campaign time over $u_*$. The colors indicate the different stability regimes in terms of $z/L$ for (a) total, (b) fine, (c) coarse, (d) super-coarse, and (e) giant dust mass concentration.

might have affected the measurements more than for the FPS. As the diameters from the analysis of the FPS differed in every analysis step and to make them comparable to the aerosol spectrometer data, the FPS were linearly interpolated and binned into 25 bins over the full size range of the FPS.

Fig. 15 presents the mass of particles deposited per mm$^2$ per day, normalized by dlogD from the FPS and the average across instruments from the aerosol spectrometers at different time steps. The samples and corresponding time frames are shown in 625    Table B1. For the aerosol spectrometer measurements, the number of particles deposited per unit area and their corresponding



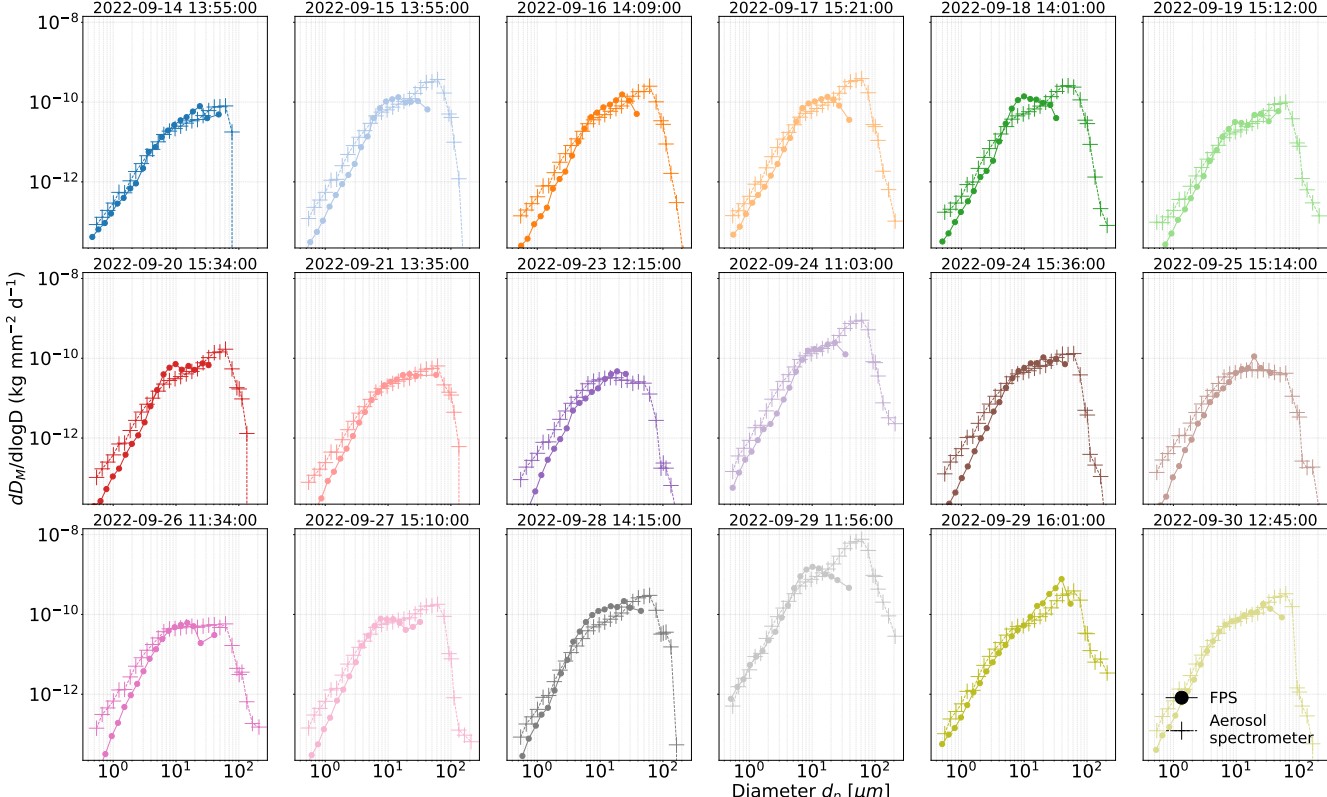

**Figure 15.** Mean aerosol spectrometer mass deposition rates from SANTRI2, Welas and Fidas with the flat-plate sampler (FPS) within the sample time steps shown one top of every subplot and in Table B1.

particle sizes were determined as outlined in Sect. 2.1.3 assuming a constant deposition velocity with particle size. The PSDs for different time steps derived from the FPS and aerosol spectrometer show reasonable agreement (always less than an order of magnitude difference, mostly less than 20% deviation). However, the PSDs sometimes differ in the position of the peak which is often at $d_p \approx 60\,\mu\text{m}$ for the aerosol spectrometer measurements but at smaller particle sizes for the FPS (e.g. 2022-09-17).

In many cases, the peak of the FPS is vague (2022-09-27) or matches the one of the aerosol spectrometer (e.g., 2022-09-21, 2022-09-24 11:03, 2022-09-26).

Overall, the agreement between the PSDs obtained from different measurement techniques is quite good, considering the differences in sampling methods and instrument principles. However, discrepancies between the PSDs may be attributed to factors such as variations in sampling efficiency, changes in wind conditions, the underlying assumption on shape and orientation

on the substrate or limitations in accurately capturing super-coarse and giant particles with the aerosol spectrometer setup, as discussed in Sect. 3.4. Despite these differences, the overall trends align well, reinforcing the robustness of the measurements across different techniques.





In a previous study (Panta et al., 2023), the aerosol spectrometer data aligned the FPS measurements more closely (almost perfectly matching, deviation for $d_p \approx 1.2\,\mu m$ with less than $10\,\%$ deviation. However, in Panta et al. (2023), although the spectrometer measurements were conducted with a Fidas 200S (included in our averaged aerosol spectrometer data), a smaller size range ($d_p = 0.2 - 19\,\mu m$) was probed and another magnification was used for the SEM. In the smaller size range, inlet inefficiencies may have a negligible effect, as these particles are less likely to experience significant losses during sampling. Additionally, the re-binning of the FPS data in those studies was coarser, which may have reduced the observed differences.

We assumed a constant deposition velocity across all size ranges, a simplification that might have not fully captured the actual deposition dynamics. Larger particles are expected to typically experience higher gravitational settling velocities, while smaller particles are more influenced by atmospheric turbulence and Brownian motion. We tested different deposition assumptions, including those appropriate for larger particles as discussed in Adebiyi et al. (2023), but found that the constant deposition velocity provided the best fit to the deposition patterns observed with the FPS. This result may suggest that the strong deposition of large particles near emission sources may be even more doubtful than calculated in Adebiyi et al. (2023) as previously discussed in earlier studies (van der Does et al., 2018; Adebiyi and Kok, 2020; Ratcliffe et al., 2024). However, as discussed in Sect. 2.1, the assumptions that the particles reach their terminal fall speed might not be applicable due to potential turbulent behavior. Additionally, it is important to consider that the FPS may tend to overestimate the abundance of larger particles if smaller particles, which follow the airflow more efficiently, are less likely to settle onto the collection substrate. This could also explain why a lower settling velocity than would be expected from Stoke's settling for large particles yields a better comparison. The assumption of a constant deposition velocity across size ranges for aerosol spectrometers with the FPS data raises questions about the actual deposition processes for super-coarse and giant particles and should be further investigated in future research.

### 3.7 Comparative analysis of J-WADI data with other field campaigns

To contextualize the findings from the current study, we compared our results with previous research on mineral dust size distributions. Formenti and Di Biagio (2024) conducted a comprehensive analysis of mineral dust aerosol size distributions, synthesizing data from more than 50 years of in situ field observations to create a harmonized dataset. They organized dust size distributions by the stage in the dust transport life cycle: source (SOURCE, within one day after emission), mid-range transport (MRT, one to four days of transport), and long-range transport (LRT, more than four days of transport). Here, we compare our J-WADI dataset with the Formenti and Di Biagio (2024) SOURCE dataset, acknowledging that their conversion to geometric diameters was not completely equal to the ones we applied.

Figure 16 compares the SOURCE data from different field campaigns Formenti and Di Biagio (2024). The mean for at least two PSDs from the Formenti and Di Biagio (2024) dataset is indicated in black, with the standard deviation in gray. The averaged J-WADI data over the whole campaign, including dusty periods and non-dusty periods, are shown in dark red. As an example for dusty conditions, we also present results during daytime (10:00-15:30 UTC) on 29 September 2022, which is depicted in dark green, with shaded areas indicating the standard deviation. The averaged J-WADI data aligns well with the averaged SOURCE dataset from Formenti and Di Biagio (2024), demonstrating overall consistency in the general shape of the




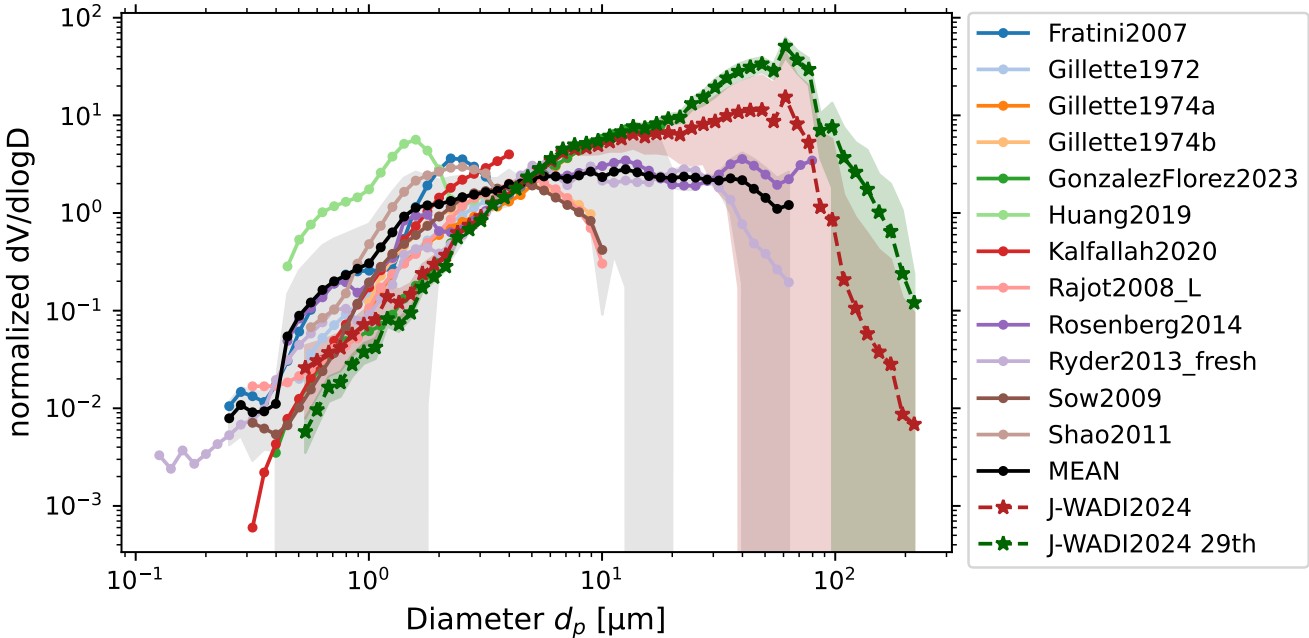

**Figure 16.** PSDs from different field campaigns within one day after emission from Formenti and Di Biagio (2024), all normalized at the integral of 1 between 1.58 and 7.1 µm before weighting by dlogD. The mean of PSDs, where at least two datasets are available in the diameter range, is indicated in black and the standard deviation in gray. J-WADI results averaged over the entire campaign are shown in dark red and for dusty conditions on September 29 in dark green with shaded areas indicating the standard deviations across time.

PSD. However, some differences are evident: For fine and coarse particles up to around 6 µm, the J-WADI PSD exhibits lower values compared to Formenti and Di Biagio (2024), while for larger particles, the J-WADI PSD is elevated and extends to even larger particle diameters. This behavior becomes more pronounced in the dataset from 29 September, where the PSD reveals a distinct enhancement in the super-coarse and giant particle size ranges. In the averaged dataset from Formenti and Di Biagio (2024) (Fig. 16, black line), no clear maximum in the volume concentration PSD is observed. In contrast, a peak around 60 µm is visible in the both J-WADI datasets.

**Table 2.** Comparison of volume size distribution percentages across different particle diameter ranges for the SOURCE dataset from Formenti and Di Biagio (2024) and from J-WADI (this study).

| Dataset | $D \leq 2.5\,\mu m$ | $2.5 < D \leq 10\,\mu m$ | $10 < D \leq 62.5\,\mu m$ | $D > 62.5\,\mu m$ |
|---|---|---|---|---|
| SOURCE (Formenti et al., 2024) | 10.8% | 34.9% | 52.7% | 1.6% |
| J-WADI complete measurement period | 0.5% | 11.9% | 59.1% | 28.5% |
| J-WADI 29 September 2022 | 0.3% | 7.8% | 57.4% | 34.5% |



Compared to SOURCE, our normalized J-WADI dataset suggests a shift in the emitted dust size distribution toward coarser particles as shown in Table 2. While Formenti and Di Biagio (2024) reported a fine particle ($< 2.5\,\mu m$) contribution of 10.8%,
our dataset shows only 0.5%, indicating a lower proportion of fine particles in our measurements. The fraction of $2.5–10\,\mu m$ particles is also smaller during J-WADi (34.9% vs. 11.9%). In contrast, our results indicate a higher proportion of super–coarse particles (59.1% vs. 52.7%) and much higher fraction of giant particles ($>62.5\,\mu m$; 1.6% vs. 28.5%).

The results from 29 September indicate lower fine (0.3%) and coarse particle fractions (7.8%). Additionally, the data show a similar contribution for the super-coarse range (57.4%) than on average for J-WADI (59.9%) and for SOURCE (52.7%).
Especially for giant particles, the fraction with 34.5% is significantly increased for the 29 September in comparison to the average J-WADI data (28.5%) and the source data (1.6%). This indicates that dust emission in J-WADI was characterized by a smaller proportion of intermediate-sized particles (fine, coarse) and a larger proportion of larger particles (super-coarse and giant) compared to the reference data.

In the 29 September J-WADI data, we included measurements taken during active dust emission. These conditions are
similar to aircraft campaigns as for the two datasets including $d_p > 20\,\mu m$ (Ryder et al., 2013; Rosenberg et al., 2014), where they actively targeted dust outflow regions and therefore tended to sample elevated dust concentrations. On the other hand, the J-WADI campaign was conducted directly at an emission source, where a higher fraction of super-coarse and giant particles is expected, as these larger particles are more likely to settle out of the atmosphere before reaching greater height and distance from the source and may explain the elevated contributions of super-coarse and giant particles in our data. This proximity to
emission sources in J-WADI may therefore explain the elevated contributions of super-coarse and giant particles in our data.

## 4   Conclusions

An in-depth understanding of the full size distribution of mineral dust at emission and its behavior during atmospheric transport is crucial for an accurate representation in climate models and for assessing dust impacts on the climate and Earth systems. Large particles remain significantly underrepresented in models, largely due to an incomplete understanding of their physical
behavior. This challenge is further compounded by the scarcity of observational data, as the measurement of large particles involves considerable technical and conceptual difficulties.

The comprehensive field measurements conducted during the Jordan Wind erosion And Dust Investigation (J-WADI) campaign have provided valuable insights into the size distribution of mineral dust particles ranging from ~0.4 to $200\,\mu m$ at a desert emission source in Wadi Rum, Jordan. This study is the first to encompass such a broad range of particle diameters directly at
the emission source, with a particular emphasis on super-coarse ($10 < d_p \leq 62.5\,\mu m$) and giant ($d_p > 62.5\,\mu m$) particles.

A key feature of this study was the utilization of a diverse set of aerosol spectrometers, including active, passive, and open-path devices, and their comparison with physical samples from a flat-plate sampler. The aerosol spectrometers covered different size ranges that were partly complementary to extend the overall observed size range, and partly overlapping to enable systematic intercomparison and validation. While agreement in mass concentrations was good for smaller particle sizes,



discrepancies arose for particles with $d_p > 10\,\mu\text{m}$, largely due to differences in measurement principles (e.g., light source and the illumination of the particles), size ranges (sensitivity limitations), and inlet effects.

Our results show that during active dust emission events, which typically occurred at friction velocities ($u_{*t}$) exceeding $0.22\,\text{m}\,\text{s}^{-1}$, 0.3 % of the mass concentration was found in the fine, 7.8 % in the coarse, 57.4 % in the super-coarse range and 34.5 % in the giant range during a dust event on 29 September 2022. Data averaged over the whole campaign (including periods

of calm winds) showed a slight shift toward the fine and coarse size fractions (0.5%, 11.9 %, 59.1 % and 28.5 %), differing from findings from previous studies as compiled in Formenti and Di Biagio (2024) by showing larger proportions in the super-coarse and giant ranges.

We found that with higher friction velocities ($u_*$) and under (near) unstable and neutral atmospheric stability conditions, dust concentrations were highest and the abundance of super-coarse and giant particles was largest. A peak in mass concentration

PSD during periods of active emission was observed at around $60\,\mu\text{m}$, although the detection of larger particles was likely constrained by inlet inefficiencies and instrument insensitivity near the limits of their size ranges. Despite this, physical samples collected using a flat-plate sampler largely confirmed the PSDs derived from aerosol spectrometers.

The results highlight the challenges in accurately quantifying giant particles but also demonstrate strategies to overcome these challenges. A better characterization of inlet dynamics is necessary to advance the measurement of (dust) aerosol PSDs

in the future, particularly for large particles. Future work should also focus on further understanding the flow dynamics of the flat-plate sampler, how particles are deposited on the substrate, and how they influence the observed PSDs, especially for capturing the largest particles, and refining methods to harmonize size distribution data from different measurement techniques. In general, especially the super-coarse and giant particle range should be measured with several instruments to cover a large size range and to eliminate differences between instrument principles. Additionally, further research should focus on the inves-

tigation of particle shape and refractive index to better link different equivalent diameters (e.g., projected area and geometric). Despite these challenges, our results demonstrate a remarkably high abundance of super-coarse and giant particles in emitted dust. This emphasizes the need to account for the full PSD, including super-coarse and giant particles, in future studies.

This study advances our understanding of emitted dust PSD variability, particularly super-coarse and giant particles. By improving our knowledge of the size distribution and abundance of these particles at emission, we lay the foundation for

unraveling their evolution during atmospheric transport and their broader impacts on the climate system. Incorporating more accurate PSDs, particularly of super-coarse and giant particles, into dust models is essential for improving predictions of long-range dust transport, cloud microphysics, and radiative forcing. These advances will ultimately enable better assessments of the environmental and climatic impacts of mineral dust.

## Appendix A: Flux density for calculating $u_{*t}$

Figure A1 shows the flux density from SANTRI4 together with $u_*$ to retrieve $u_{*t}$ as described in Sect. 2.1.1.





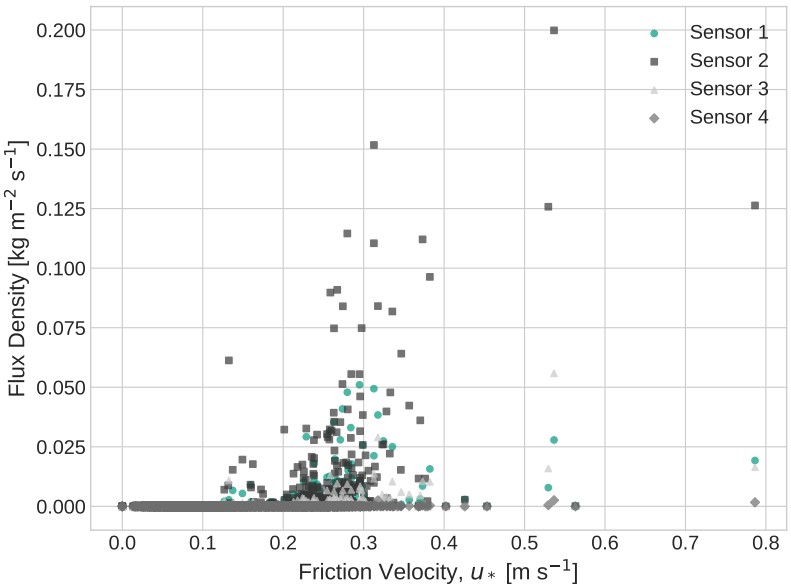

**Figure A1.** Flux density for the four sensors of the SANTRI4 over $u_*$ to retrieve $u_{*t}$.

**Table B1.** FPS and their corresponding time frames.

| sample code | start timecode (UTC) | stop timecode (UTC) | sample code | start timecode (UTC) | stop timecode (UTC) |
|---|---|---|---|---|---|
| WRS_01 | 12.09.2022 13:55 | 13.09.2022 13:55 | WRS_12 | 24.09.2022 11:03 | 24.09.2022 15:36 |
| WRS_02 | 13.09.2022 13:55 | 14.09.2022 13:55 | WRS_13 | 24.09.2022 15:36 | 25.09.2022 15:14 |
| WRS_03 | 14.09.2022 13:55 | 15.09.2022 13:55 | WRS_14 | 25.09.2022 15:14 | 26.09.2022 11:32 |
| WRS_04 | 15.09.2022 13:55 | 16.09.2022 14:09 | WRS_15 | 26.09.2022 11:34 | 27.09.2022 15:02 |
| WRS_05 | 16.09.2022 14:09 | 17.09.2022 15:21 | WRS_16 | 27.09.2022 15:10 | 28.09.2022 14:15 |
| WRS_06 | 17.09.2022 15:21 | 18.09.2022 14:01 | WRS_17 | 28.09.2022 14:15 | 29.09.2022 11:56 |
| WRS_07 | 18.09.2022 14:01 | 19.09.2022 15:12 | WRS_18 | 29.09.2022 11:56 | 29.09.2022 16:01 |
| WRS_08 | 19.09.2022 15:12 | 20.09.2022 15:34 | WRS_19 | 29.09.2022 16:01 | 30.09.2022 12:45 |
| WRS_09 | 20.09.2022 15:34 | 21.09.2022 13:35 | WRS_20 | 30.09.2022 12:45 | 01.10.2022 15:54 |
| WRS_10 | 21.09.2022 13:35 | 23.09.2022 12:15 | WRS_21 | 01.10.2022 15:54 | 02.10.2022 15:23 |
| WRS_11 | 23.09.2022 12:15 | 24.09.2022 11:03 | | | |

## Appendix B:  Time step information on the sampling periods of the FPS samples

Table B1 shows the sample time for each FPS sample as collected in the field. The corresponding mass deposition fluxes are shown in Figure 15.





## Appendix C: Outlier correction

Outliers were identified and removed in the SANTRI2, UCASS, and, to a lesser extent, in the Welas and CDA data as described in the following.

### C1 SANTRI2

Since the SANTRI2 is an open-path instrument, its sensors are directly exposed to environmental factors such as sunlight, shadows, light reflections from nearby metal, and dirt. This exposure can result in artifacts in the data counts in different bins.
Additionally, fluctuations in light source intensity may be misinterpreted by the sensors as particles in the air. Both SANTRI2 upfacing units (2U and 4U) exhibited more daytime peaks than nighttime peaks during J-WADI and generally reported fewer counts compared to SANTRI2_2D and 4D, which displayed less systematic behavior with more disordered and ubiquitous high peaks and more counts in general. The behavior of the upfacing units (SANTRI2_xU) reflects the observations made by other instruments that dust concentrations during nighttime were generally lower due to calm winds. To better understand this
behavior, we investigated whether the elevated counts in the downward-facing SANTRI2 units were due to their orientation toward the light. Consequently, between 13:08 UTC on September 19 and 16:30 UTC on September 21, we inverted the upfacing units for testing so that they faced downward. No direct correlation was observed between turning the unit and corresponding counts in 2U and 2D. During this period, counts in SANTRI2 2D changed but not immediately after turning the unit (16:08 local time = 13:08 UTC) and persisted even after turning the unit back, suggesting no significant impact. With
increasing bin size, outliers became less pronounced, with most outliers disappearing in the 99 – 125 μm bin. This reduction in outliers may stem from higher noise sensitivity in smaller bins, where the corresponding voltage levels are relatively low. Although noise decreased for larger bins, the downward-facing units still show numerous unrealistic outliers, especially in the smaller bins, and also unrealistically high counts in the larger bins. Consequently, we decided to exclude the downward-facing units from our analysis, as these outliers do not appear to arise from misdirected light reflections but rather from other hardware
or software issues.

For the remaining two units, we applied the following steps: (1) times periods of instrument cleaning were removed from the data entirely; (2) we excluded sensors across all bins for the time frames which included highly unrealistic counts exceeding on average four times those recorded by other sensors; (3) Sensor 1 of SANTRI2_4U was removed entirely from the dataset due to persistent unrealistic behavior. After these initial steps, additional outliers were identified and removed using a statistical
comparison with measurements of other sensors of the same and of other SANTRI2 devices. The comparison was applied to all data points at their original 1 Hz frequency as discussed below:

**I: Intra-Sensor and Intra-Instrument Comparison (Bins and Flux)**

1. Signal 1 recorded by SANTRI2 corresponds to the voltage obtained by the photosensor. We removed counts which were outside the signal range 2750 to 3250 mV.





2. Signal 2 is the indicator of IR-led light source intensity required to keep the detector (Signal 1) in the range from 2500 - 3500 mV. We did not see any abnormal behavior here for the remaining two units.

3. The flux recorded by SANTRI2 is proportional to the cross-sectional area of the particles (Flux 1 over all bins, Flux 2 only the upper 3 bins) of all particles recorded in the corresponding time. Based on the visual observation of the time series, we assume that a Flux 1 value exceeding 1000 is unrealistic. Therefore, data points for which Flux 1 surpassed this threshold were removed.

4. For a given time step in which a count in the smallest bin was significantly higher than its surrounding counts and which exceeded a predefined threshold, the counts in all bins were set to NaN. We considered these counts outliers, as observing numerous giant particles in one second and much fewer in the next was deemed unrealistic. This procedure was applied using two methods:

    (a) **Local Anomaly in Individual Counts**: An individual count was considered an outlier if it was at least four times higher than the sum of the surrounding 180 data points. This method aimed to identify sharp, localized spikes in the data that significantly deviated from their immediate surroundings.

    (b) **Localized Spike in Sum**: An outlier was flagged if the sum of the surrounding 30 data points was at least ten times higher than the sum of the surrounding 300 data points. Additionally, individual counts were only removed if their value exceeded 2, to prevent the removal of valid low-magnitude data.

**II: Inter-Sensor and Inter-Instrument Comparison**

1. We averaged counts over 3600 points (1 hour) and compare the 73-80 µm bin (first bin) to the mean of all other time-averaged sensors. If the mean of the 73-80 µm bin was greater than 3 counts (in order to not remove single counts/arbitrary threshold) and 10 times higher than the averaged other counts, we set the corresponding count in all bins to NaN.

The thresholds and time windows for these methods were chosen based on exploratory analysis, as there is no defined standard for identifying outliers in data for such large particles. While somewhat arbitrary, these parameters effectively removed unrealistic spikes without eliminating data we considered realistic based on observatory analysis and in comparison with other instruments.

## C2  UCASS

Although the electronics of the UCASS were protected from ambient light by a housing (Fig. 5), some light could still enter through the cylindrical opening but this did not result in systematic outliers. However, we observed a large number of counts with recurring values, such as 243, 512, 514, and 65535 ($2^{16} - 1$) but also others. The value 65535 corresponds to $2^{16} - 1$, which is the maximum value that can be recorded using a 16 bit variable. However, since the UCASS records data using 12-bit variables, the occurrence of this value indicates an anomaly, specifically suggesting faulty communication between the





UCASS and the Raspberry Pi. Those recurring values were removed from the data. For the remaining data, to identify other recurring count values, first, for each instrument, a count distribution of detected values was created for all particle size bins. The maximum observed value in each column was determined, and a complete range of integer values up to this maximum was generated. The actual frequency of occurrence for each integer value was then compared against this range, with missing

values set to zero. To identify outliers, each detected value was assessed based on its local neighborhood. Specifically, for each value, the sum of occurrences in the five preceding and following integer bins was computed. If this sum was less than one-tenth of the observed value, the value was flagged as an outlier. Once the outliers were identified, a thresholding step was applied to remove only those outliers exceeding a minimum value of 2 to reduce noise in the filtering process. These flagged values were replaced with NaN values. Consequently, we excluded these values from the analysis.

**C3   Welas, Fidas, and CDA outlier correction**

There were almost no detectable outliers in Welas, Fidas, and CDA, except that occasionally, they measured counts for large $d_p$, but not in the smaller sizes. For instance, occasionally, they measured ten counts in a size range $d_p > 60\,\mu\mathrm{m}$ but none for $40\,\mu\mathrm{m} < d_p < 60\,\mu\mathrm{m}$ in a 15 minutes time interval. For Welas and CDA, we applied a filtering criterion for particles larger than $10\,\mu\mathrm{m}$, where we removed counts if particles were detected in one size bin but not in the preceding, smaller size bin.

This criterion was applied to the integrated size distribution, where approximately eight raw bins were combined, over the 15-minute averaging intervals. Such outliers could arise due to inlet inefficiencies, which can lead to inaccurate size measurements, inconsistent particle sampling, or wrongly interpreted light scattering.

**Appendix D:   Intercomparison and bias correction**

To identify and correct systematic errors between individual instruments, we conducted an intercomparison. From October 2nd

to October 5th, 2022, at the end of the J-WADI measurement period, Welas, Fidas, CDA, and UCASS were installed in close proximity to each other at 2 m height for comparative analysis. The SANTRI2 units were mounted also next to each other on the ground and vertically, i.e. in their standard setup, to capture more large particles, now transported in saltation, and thereby to obtain a more robust statistical comparison. Unfortunately, there were no notable dust events during this period, which posed limitations to the comparative assessment, but we still measured particles in the time frame. In Sects. D1 – D2, we describe

three procedures applied to correct for systematic errors.

**D1   Systematic error correction via linear regression of Welas, Fidas, and SANTRI2**

To remove systematic biases in dust concentration measurement between aerosol spectrometers of the same type (here: Welas, Fidas, and SANTRI2), a similar approach to the method described by González-Flórez et al. (2023) and Dupont et al. (2024) was applied. The average dust concentration in each 15-minute bin from one instrument was compared to the corresponding

values from the other aerosol spectrometer of the same type. The systematic correction parameter, $\lambda_i$, for each bin $i$, was



calculated as the slope of the regression between the concentrations of the compared instrument bins:

$$c_{oc}(d_i) = \lambda_i c_r(d_i), \tag{D1}$$

where $d_i$ represents diameter of bin $i$, $c_r$ the concentration from the reference instrument and $c_{oc}$ the concentration from the instrument to be corrected. A $\lambda_i > 1$ indicates that the concentration of the reference instrument was lower, and $\lambda_i < 1$ indicates that the concentration of the reference instrument was higher. A perfect match would yield a correction factor of one. The corrected concentration ($c_{om}$) was then obtained as:

$$c_{oc}(d_i) = c_{oc}^{\text{uncorr.}}(d_i)/\lambda_i. \tag{D2}$$

The correction parameter obtained during the intercomparison period was applied to the entire measurement period. The Pearson correlation coefficient $r$ was used to assess the correlation between the instruments. At correlations less than $r = 0.6$, no correction was applied. For the Welas, few data points existed above $d_p > 45\,\mu m$ with low mass concentrations with many values being 0 in one instrument and small number in the other, therefore the concentrations across the last bins ($d_p > 45\,\mu m$) were averaged and treated together. This procedure was implemented for the different instrument types. Here, Fidas_4m, Welas_4m, and SANTRI2_2 m served as the reference, while Fidas_2m, Welas_2m, and SANTRI2_4 m were corrected and adjusted by the slope determined from the linear regression. The issue of unrealistically large numbers of counts in some bins of the SANTRI2 persisted during the intercomparison period for the SANTRI2s. For correction, we applied the first step of the outlier correction method (Sect. C1) without adjusting for higher fluxes, as higher fluxes are possible due to SANTRI2 being positioned at ground level.

$\lambda_i$ for the different instruments and the corresponding bins or bin groups is given in Table D1, D3, and D2. For SANTRI2, the sensors closer to the ground (ascending from S1 to S5) and for small bins $\lambda_{i,S}$ are closer to 1, decreasing for sensors in more distance to the ground and larger bins. For the largest bins and the sensors furthest from the ground the correlation was $< 0.6$, so $\lambda_{i,S}$ was set to 1.000. For Welas and Fidas, the values were closer to 1, except for Welas $d_i > 45\,\mu m$ (= 0.5) and $d_i = 1\,\mu m$ (= 0.1). For the last three bins of the Fidas, correlation was also $< 0.6$, no correction was applied (indicated by 1.0 in Table D3).

**Table D1.** Correction parameter $\lambda_{i,S}$ to correct SANTRI2_4U. $\lambda_{i,S}$ was set to 1, if the correlation was lower than 0.6.

| Sensor | Bin1 | Bin2 | Bin3 | Bin4 | Bin5 | Bin6 | Bin7 |
|---|---|---|---|---|---|---|---|
| S1 | 0.9706 | 0.9720 | 0.9745 | 0.9722 | 0.9643 | 0.9739 | 0.9925 |
| S2 | 1.0493 | 1.0349 | 1.0548 | 1.0548 | 1.1372 | 1.1793 | 1.2041 |
| S3 | 1.1832 | 1.0386 | 0.8594 | 0.8481 | 0.7644 | 0.6887 | 1.0000 |
| S4 | 0.7876 | 0.6137 | 0.4290 | 0.3004 | 1.0000 | 1.0000 | 1.0000 |
| S5 | 0.4518 | 0.4193 | 0.4217 | 0.1936 | 1.0000 | 1.0000 | 1.0000 |





**Table D2.** Correction parameter $\lambda_{i,W}$ to correct Welas_2m. $\lambda_{i,W}$ was set to 1, if the correlation was lower than 0.6.

| $d_i$ in µm | 1.5 | 1.8 | 2.1 | 2.6 | 3.1 | 3.7 | 4.5 | 5.5 | 6.6 | 8 | 9.6 | 12 | 14 | 17 | 21 | 26 | 31 | 38 | >45 |
|---|---|---|---|---|---|---|---|---|---|---|---|---|---|---|---|---|---|---|---|
| $\lambda_i$ | 0.1 | 0.6 | 0.8 | 0.7 | 0.8 | 0.8 | 0.8 | 0.7 | 0.6 | 0.7 | 0.7 | 0.7 | 0.7 | 0.7 | 0.7 | 0.7 | 0.7 | 0.6 | 0.5 |

**Table D3.** Correction parameter $\lambda_{i,F}$ to correct Fidas_2m. $\lambda_{i,F}$ was set to 1, if the correlation was lower than 0.6.

| $d_i$ in µm | 0.36 | 0.49 | 0.67 | 0.92 | 1.2 | 1.7 | 2.4 | 3.4 | 4.7 | 6.4 | 8.8 | 12 | 16 | 22 | 30 | 40 |
|---|---|---|---|---|---|---|---|---|---|---|---|---|---|---|---|---|
| $\lambda_i$ | 0.9 | 1 | 0.9 | 1 | 1 | 0.9 | 0.8 | 0.8 | 0.8 | 0.8 | 0.9 | 0.7 | 0.8 | 1.0 | 1.0 | 1.0 |

### D2 Systematic instrument differences correction via comparison with Fidas_4m

860 Since the instruments rely on slightly different measurement principles, harmonizing their outputs is essential to ensure consistent and comparable PSD data. This harmonization minimizes systematic biases and allows for intercomparison across instruments.

In this study, we use Fidas_4m as the reference instrument due to its reliable performance and broad operational size range, which overlaps to some extent with all other instruments except with the SANTRI2. This makes it well-suited for establishing 865 correction factors for the other instruments. To remove systematic differences between instruments, we applied a constant correction factor relative to the reference instrument after re-binning all instruments to the Fidas bins, as explained in Sect. F. The correction factor for each instrument was obtained by minimizing the difference between its PSD and that of the Fidas_4m during the intercomparison period. The correction factors were determined in specific size ranges in which they overlapped with Fidas_4m and which were not too close to the limits of their measurement range. For the Welas, a size range of $2-7\,\mu m$ 870 was used, while size ranges of $2-10\,\mu m$ and $4-7\,\mu m$ were applied for UCASS A and UCASS B, respectively. For the CDA, a size range of $5-10\,\mu m$ was used (CDA and both UCASSs not used for the later analysis).

To ensure the statistical robustness of the correction, we calculated the correction factor for every 15 minute time step in the intercomparison period and then we used a trimmed mean (5% on every end) over the whole time frame to calculate the scaling factors, reducing the influence of outliers. The scaling factors obtained were 1.03 for UCASS A, 1.22 for UCASS B, 875 1.10 for Welas_4m, 1.15 for Welas_2m, and 0.62 for CDA.

### D3 Systematic x-axis (diameter) correction of the Welas

During the campaign, the Welas lamps were not exchanged. We assume that degradation of the lamps due to their limited nominal lifetime of approximately 400 hours, may have led to a gradual shift in bin classification toward smaller bins over time. Such an effect was evident in the Welas_2m data, as its results in number size distribution showed a large deviation in number size distribution to the Welas_4m and to other instruments. For instance, it mostly recorded smaller concentrations 880 than Welas_4m (contrary to what we expected) and smaller than Fidas_2m. In addition, it mostly measured less large particles than Welas_4m during the campaign (Fig. 9a). To correct this bias, we sought a method to transform the x-axis (diameter) of





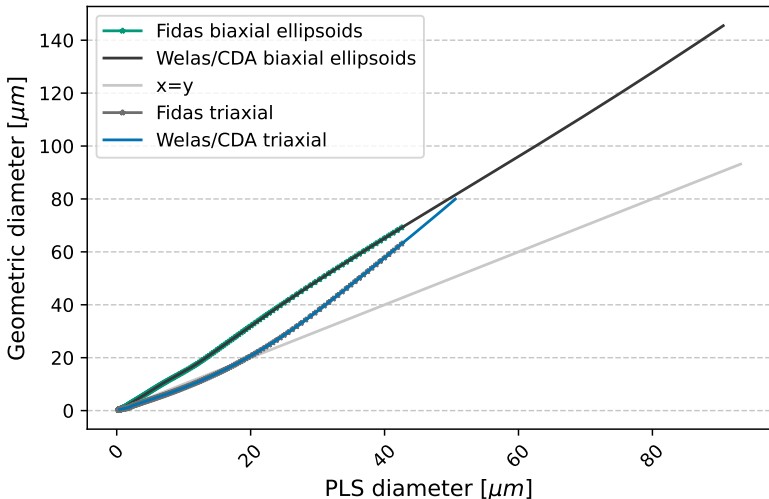

**Figure E1.** Conversion from optical (retrieval for PLS, x-axis) to geometric diameter (biaxial spheroids, y-axis) for Welas, Fidas, and CDA.

the PSDs by analyzing data collected during the intercomparison period, where both Welas instruments measured next to each other at the same height.

From this analysis, we identified a quadratic relationship of the form $a + bx + cx^2$ to correct the bin boundaries, with the parameters $a = 0.16$, $b = 1.15$, and $c = 0.006$. This correction approach was implemented by optimizing the diameter-wise alignment of the size distributions using data from both instruments. The optimization minimized the discrepancy between the two instruments by comparing the linearly interpolated concentration of Welas_2m with the observed values of Welas_4m across their overlapping size ranges. The correction parameters were determined using a global optimization routine, specifi-

cally the differential evolution algorithm (SciPy's differential evolution algorithm). The correction function $a + bx + cx^2$ with the corresponding parameters were then applied to the original bin diameters of Welas_2m.

**Appendix E:  Conversion from optical to geometric diameter**

By deriving the geometric diameter for Welas, CDA, and Fidas, assuming biaxial ellipsoids as explained in Sect. 2.1.2, we made the measurements more tailored to dust. In Fig. E1, we compare the resulting geometric diameters (1) assuming biaxial

ellipsoids with an aspect ratio of 1.49 (y-axis) and (2) assuming triaxial ellipsoids with the default optical diameters from PSL (x-axis). This shows that optical diameters of PSL tend to underestimate the sizes of biaxial dust particles, particularly for larger particles. In contrast, the transformation for smaller particles is minimal, which can be attributed to the combined effects of dust asphericity and the refractive index.

To determine the ratio of the volume-equivalent diameter $d_{geo}$ to the projected-area diameter $d_{\text{PA}}$ for the SANTRI2, the

following equation was used, assuming randomly oriented particles:



$$d_{geo} = d_{\text{PA}} \cdot \sqrt{AR \cdot HWR} \cdot \left( \frac{3}{AR^p + HWR^p + (AR \cdot HWR)^p} \right)^{\frac{1}{2p}}, \qquad (\text{E1})$$

with, $AR$, the aspect ratio (length-to-width) and $HWR$ is the height-to-width ratio, and $p$ is a shape-dependent exponent:

$$p = \frac{\log(3)}{\log(2)}. \qquad (\text{E2})$$

This equation, derived from the surface area of spheres and triaxial ellipsoids, provides a geometric approximation of the relationship between $d_{geo}$ and $d_{\text{PA}}$. While biaxial spheroids were assumed for the main analysis (consistent with Fidas and Welas), a more precise analysis should consider triaxial ellipsoids with $HWR \neq 1$ (Huang et al., 2020). We did use this approach in order to be in line with the assumptions for Welas and Fidas and the database to convert their optical diameters did not provide a size range $> 50\,\mu\text{m}$ (PSL, Meng et al., 2010). The results for triaxial ellipsoids are briefly discussed in Appendix H.

For the SANTRI2, $AR = 1.49$ (derived as median AR from SEM analyses) and $HWR = 1$ (in line with the biaxial assumption of the Welas and Fidas), the computed ratio is:

$$\frac{d_{geo}}{d_{\text{PA}}} = \sqrt{1.49} \cdot \left( \frac{3}{1.49^p + 1^p + (1.49)^p} \right)^{\frac{1}{2p}} \approx 1.055. \qquad (\text{E3})$$

For the SEM analyses, we used the approach of Huang et al. (2021). Their assumption is that the FPS collects dust particles in an orientation where their largest surface lies parallel to the collection substrate, meaning the smallest dimension (height $H$) is aligned perpendicular to the surface (Huang et al., 2021). Here, $HWR = 1$ but AR was used directly from the SEM analysis for every diameter.

$$\frac{d_{area}}{d_{geo}} = \frac{\sqrt{LW}}{\sqrt[3]{LWH}} = \frac{\sqrt[6]{AR}}{\sqrt[3]{HWR}}. \qquad (\text{E4})$$

**Appendix F: Re-binning method for harmonizing and comparing PSD**

To harmonize the PSDs from different instruments and enable averaging into a single PSD, we applied a re-binning method that calculates bin-weighted averages. The instruments included in this process are the two Welas, two Fidas, and the upward-looking units SANTRI2_xU. Due to differences in the operational size ranges of these instruments, we included their data to obtain an average PSD only above specific size thresholds to avoid inaccuracies near the boundaries of their measurement ranges. For the Welas, data were included from $d_p > 1.5\,\mu\text{m}$, for the Fidas from $d_p > 0.5\,\mu\text{m}$, and for the SANTRI_xU from $d_p > 80\,\mu\text{m}$.

The re-binning method interpolates measurements from the original (old) bin edges to a common set of target (new) bin edges. For each time step, we calculated the contributions of the old bins to the new bins by determining the overlapping bin widths. The relative amount from the old bin that overlaps with the new bin was than included in the new bin. This approach





accounts for both constant and varying (such as for SANTRI2) bin edges over time. In cases where no overlap exists between old and new bins, or where contributions to a specific new bin are entirely invalid (e.g., NaN values), the resulting value is set

to NaN.

## Appendix G:  Sampling efficiency

Inlets are critical components of instruments used for aerosol sampling, and they serve the purpose of guiding particles into the measurement system. Aspirating an aerosol sample through an inlet toward a sensor involves several complexities. The sizes of the particles, inlet design as well as wind conditions in comparison to the aspiration airstream inside the inlet determine the

occurring particle losses. The sampling efficiency $\eta_{\text{sampling}}$ is used to describe the effectiveness of an inlet system in capturing and transporting particles from the environment to the measurement chamber. It is defined as the product of inlet and transport efficiencies, $\eta_{\text{inlet}}$ (draw particles into an inlet) and $\eta_{\text{transport}}$ (losses inside the inlet):

$$\eta_{\text{sampling}} = \eta_{\text{inlet}} * \eta_{\text{transport}}. \tag{G1}$$

When measuring larger particles, such as (super-) coarse and giant mineral dust particles, the choice of inlets can significantly

impact the amount of particles detected and is therefore discussed for the different instruments used in this study.

## G1    Inlet efficiency

The inlet efficiency ($\eta_{\text{inlet}}$) is the product of the aspiration and transmission efficiencies, representing the fraction of ambient particles delivered to the sampling system, defined as:

$$\eta_{\text{inlet}} = \eta_{\text{asp}} * \eta_{\text{transm}}. \tag{G2}$$

For instruments that actively aspirate air with pumps, i.e., the directional and the Sigma-2 inlet in our study, measurements suffer from anisokinetic conditions which alter the particle concentration in the nozzle of the inlets compared to the original samples. Super-isokinetic conditions, where the airflow speed in the inlet $U$ exceeds ambient air speed $U_0$, lead to an underestimation of larger particles. Sub-isokinetic conditions, where the inlet airflow speed $U$ is slower than the surrounding air speed $U_0$, lead to an overestimation of larger particles. The concentration of particles of given size entering the inlet divided by their

concentration in the ambient environment is defined as aspiration efficiency ($\eta_{\text{asp}}$).

Several studies have investigated sampling from a flowing gas using thin-walled nozzles under various conditions, including isokinetic and anisokinetic sampling in both isoaxial and anisoaxial flows. Summaries of this work are provided by Kulkarni et al. (2011). The developed models are applied under conditions of constant ambient and sample gas velocities, which are typically much higher than the particle settling velocity, making gravitational effects negligible. Among the various studies,

we chose the correlation from Liu et al. (1989) and Zhang and Liu (1989) for the aspiration efficiency ($\eta_{asp}$) due to its applicability over a larger Stokes number (Stk) range ($0.01 \leq \text{Stk} \leq 100$) and wind speed range ($0.1 \leq \frac{U_0}{U} \leq 10$) to represent





particle sampling under various flow conditions. Aspiration efficiency ($\eta_{asp}$) is estimated as:

$$\eta_{\mathrm{asp}} = \begin{cases} 1 + \dfrac{\left[\frac{U_0}{U} - 1\right]}{\left[1 + \frac{0.418}{\mathrm{Stk}}\right]} & \text{for } \frac{U_0}{U} > 1 \\[3mm] 1 & \text{for } \frac{U_0}{U} = 1 \\[3mm] 1 + \dfrac{\left[\frac{U_0}{U} - 1\right]}{\left[1 + \frac{0.506\sqrt{U_0/U}}{\mathrm{Stk}}\right]} & \text{for } \frac{U_0}{U} < 1. \end{cases} \tag{G3}$$

The transmission efficiency ($\eta_{\mathrm{trans}}$) is the fraction of aspirated particles transmitted through the inlet (Kulkarni et al., 2011).

Inertial transmission losses have been studied by Liu et al. (1989) and Hangal and Willeke (1990). For sub-isokinetic sampling (when $\frac{U_0}{U} > 1$), particles are often deposited on the nozzle walls, resulting in a transmission efficiency of less than 1 (Liu et al., 1989). Liu et al. (1989) proposed for the inertial transmission efficiency $\eta_{\mathrm{trans,inert}}$ for sub-isokinetic isoaxial sampling as:

$$\eta_{\mathrm{transm,inert,L89}} = \frac{1 + \left[\frac{U_0}{U} - 1\right] \Big/ \left[1 + \frac{2.66}{\mathrm{Stk}^{2/3}}\right]}{1 + \left[\frac{U_0}{U} - 1\right] \Big/ \left[1 + \frac{0.418}{\mathrm{Stk}}\right]}. \tag{G4}$$

Conversely, Hangal and Willeke (1990) assume no inertial transmission losses for sub-isokinetic isoaxial sampling. For super-
isokinetic sampling (when $\frac{U_0}{U} < 1$), Liu et al. (1989) stated that the particle movement is not directed toward the walls, leading to a transmission efficiency of 1.

However, Hangal and Willeke (1990) argue that under these conditions, flow separation occurs at the nozzle inlet, leading to the formation of a constricted jet (vena contracta). This induces turbulence, which enhances particle deposition. They provide the following inertial transmission efficiency for super-isokinetic sampling:

$$\eta_{\mathrm{transm,inert,\ HW90}} = \exp\left[-75 I_v^2\right], \tag{G5}$$

where the parameter $I_v$ describes the inertial losses in the vena contracta and is given by:

$$I_v = 0.09 \left(\frac{\mathrm{Stk}}{(U_0/U)^{0.3}}\right). \tag{G6}$$

Equation (G4) applies for $0.01 \leq \mathrm{Stk} \leq 100$ and $1 \leq \frac{U_0}{U} \leq 10$ (Liu et al., 1989) and Eq. (G5) for $0.02 \leq \mathrm{Stk} \leq 4$ and $0.25 \leq \frac{U_0}{U} \leq 1.0$ (Hangal and Willeke, 1990). In our study, we combine the expression for sub-isokinetic sampling conditions from
Liu et al. (1989) and for super-isokinetic sampling from Hangal and Willeke (1990) to estimate $\eta_{\mathrm{transm,inert}}$.

We did not calculate the gravitational settling transmission efficiency $\eta_{trans,grav}$ due to the complexities and assumptions involved in accurately modeling particle deposition at the inlet, as highlighted by Kulkarni et al. (2011). Another minor issue is that the approximations might not be applicable for calm ($U_0 < 0.5\,\mathrm{m\,s^{-1}}$) and slow wind conditions $0.5 < U_0 < 1.5\,\mathrm{m\,s^{-1}}$ due to an enhanced gravitational force (Kulkarni et al., 2011). We neglect this issue here as we are particularly interested in the
large-size of the PSD, which we assume to be most relevant under higher wind speeds.

## G2    Transport efficiency

Transporting the aerosol sample through pipes to the measurement chamber involves bends and other flow elements, with either laminar or turbulent flow. Particle deposition during transport in the pipes can alter aerosol characteristics, influenced



by mechanisms such as agglomeration and re-entrainment. These phenomena depend on flow regime, rate, tube size and orientation, temperature gradients, and particle size (Kulkarni et al., 2011). The transport efficiency ($\eta_{\text{transport}}$) for a given particle size is the product of the efficiencies for each deposition mechanism, $m$, in each flow element, $f$:

$$\eta_{\text{transport}} = \prod_f \prod_m \eta_{f,m}$$

The different mechanisms and flow elements which lead to inlet inefficiencies for the inlets considered in our study are explained in the following.

Fuchs (1964) and Thomas (1958) developed expressions for gravitational settling in horizontal tubes with laminar flow. These losses are especially important for the directional inlet due to its horizontal first part. Heyder and Gebhart (1977) extended this to inclined tubes, providing a general correlation for gravitational deposition ($\eta_{\text{grav}}$) from laminar flow in circular tubes:

$$\eta_{\text{grav}} = 1 - \frac{2}{\pi}\left[2\kappa\sqrt{1-\kappa^{2/3}} - \kappa^{1/3}\sqrt{1-\kappa^{2/3}} + \arcsin(\kappa^{1/3})\right] \tag{G7}$$

with $\kappa = \varepsilon \cos\theta = \frac{3}{4}\frac{L}{d}\frac{V_{\text{ts}}}{U}\cos\theta$ and $\varepsilon = \frac{3}{4}Z = \frac{3}{4}\frac{L}{d}\frac{V_{\text{ts}}}{U}$ where $\frac{V_{\text{ts}}\sin\theta}{U} \ll 1$ with $V_{\text{ts}}$ being the settling velocity, $L$ the length of the pipe element, $\theta$ the possible inclination of the element, and $d$ the diameter. This formula applies to various tube orientations and is consistent with experimental results (Kulkarni et al., 2011). It reduces to the case of the horizontal tube when $\theta = 0°$. In vertical tubes, the transport efficiency for gravitational settling is 1 (100%) as particles do not deposit horizontally. Here, $V_{ts}$ is the settling velocity that can be approximated by Stokes settling for small particles. However, as discussed in Adebiyi et al. (2023), this cannot be applied for larger particles. Adebiyi et al. summarize alternative calculations for $V_{ts}$ and we implement Wu and Wang (2006) with a Corey Shape Factors of $= 0.7$ (typical for mineral dust).

Another relevant loss of particles during sampling is the transport efficiency for bends $\eta_{bend}$. This is an important part in the directional inlet. We implement a formula based on the experimental work developed in Wang et al. (2024) to calculate $\eta_{bend}$, as it shows advantages compared to other experimental data (Pui et al., 1987) and models proposed before Pui et al. (e.g., 1987) (Dean numbers 1000-4189, inner diameters 5-15 mm, and curvature ratios 2-10, it relies solely on the Stokes number, and accurately predicts particle transport efficiency for Stokes numbers between 0.001 and 10). It is defined as:

$$\eta_{\text{bend}} = \frac{1}{1 + (\text{Stk}/0.17)^{2.73}}. \tag{G8}$$

Turbulent inertial deposition occurs when large particles, due to their high inertia, cannot follow the curved streamlines of turbulent eddies and are deposited on the walls of a tube. For our study, this is relevant for the UCASS for their large pipe diameter but not for the directional inlet as we assume a laminar flow. This phenomenon is described by the transport efficiency $\eta_{\text{turb-inert}}$ (Kulkarni et al., 2011):

$$\eta_{\text{tube, turb-inert}} = \exp\left(-\frac{\pi d L V_t}{Q}\right). \tag{G9}$$

where $d$ is the tube diameter, $L$ is the transport length, $V_t$ is the turbulent inertial deposition velocity, and $Q$ is the volumetric flow rate. Liu and Agarwal (1974) found that the dimensionless deposition velocity ($V^+$) increases with particle relaxation





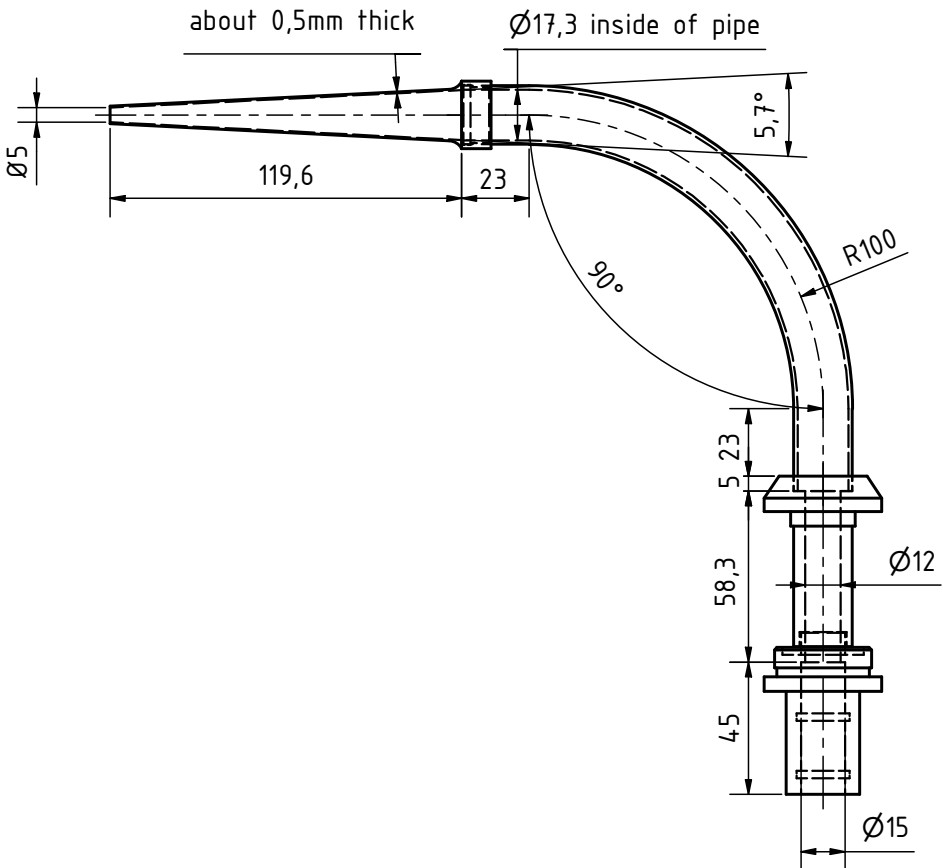

**Figure G1.** Design of the directional inlet used for Welas and Fidas. Figure by S. Haaß.

1015    time ($t^+$) in the so-called turbulent diffusion–eddy impaction regime and peaks at $t^+ \approx 30$. For larger $t^+$, deposition velocity decreases in the particle inertia-moderated regime, as reduced turbulence influence allows particles to penetrate the sublayer and deposit directly onto the wall.

The re-entrainment of particles could significantly influence the PSD. To our knowledge, there is no widely accepted approximation for the re-entrainment of particles (Kulkarni et al., 2011). We assume that especially for changing wind conditions

1020    or a moving directional inlet (Welas, Fidas, or rotating mast), re-entrainment could be a relevant alteration mechanism of the PSDs but could not be quantified further. Diffusional deposition becomes relevant only for particles $< 0.1\mu m$, and therefore we neglected it here. In addition, the calculation of the pipe enlargement was not included due to inconsistencies in published methods and difficulties in reproducing the reported calculations (Schade et al., 2007). In the following, the efficiencies of the inlets and instruments used in this study are quantitatively or qualitatively analyzed.



## G3    Directional inlet of Welas and Fidas

Welas and Fidas sampled the same air stream of $4.8\,\mathrm{l\,min^{-1}}$ and used the same inlet for guiding the aerosol to their corresponding measurement chambers during J-WADI. There, we deployed a directional inlet as shown in Fig. 6a and detailed in Fig. G1. In the directional inlet, particles are transported through a horizontal nozzle where the inlet undergoes enlargement, through a bend, guided through a vertical pipe piece, an IADS (Intelligent Aerosol Drying System) drying system, i.e., vertical transport, and eventually reach the measurement chamber.

We calculated the sampling efficiency with the following formula:

$$\eta_{\mathrm{sampling}} = \eta_{\mathrm{inlet}} * \eta_{\mathrm{transport}} = \eta_{\mathrm{asp}} * \eta_{\mathrm{transm}} * \eta_{\mathrm{grav}} * \eta_{\mathrm{bend}}. \tag{G10}$$

For $\eta_{\mathrm{inlet}}$, we used the formulas introduced above for $\eta_{\mathrm{transm}}$ and $\eta_{\mathrm{asp}}$ (Eqs. G3, G4/sub-isokinetic conditions, G5/super-isokinetic conditions). As the Reynolds numbers for all parts of the inlet are smaller than 600, we assumed a laminar flow and that turbulent inertial deposition is negligible. The most relevant losses for the directional inlet for large particles are sedimentation losses ($\eta_{\mathrm{grav}}$) in the horizontal part of the inlet, as they tend to settle out of the airstream due to gravity. These losses can result in underestimations of larger particle concentration.

The sampling efficiency $\eta_{\mathrm{sampling}}$ efficiency for the directional inlet is shown in Fig. 10a for different wind conditions $U_0$. The dotted lines indicate that one or more formulas discussed above are not proven for the conditions shown in the figure. For wind speeds $U_0 \leq 5\ \mathrm{m\,s^{-1}}$ and particle diameters $d_p \leq 5\,\mu\mathrm{m}$, the efficiency $\eta_{\mathrm{sampling}} \approx 100\ \%$ and decreases to 0 % at $d_p \approx$ $30\,\mu\mathrm{m}$. For $U_0 > 5\ \mathrm{m\,s^{-1}}$ and $d_p > 5\,\mu\mathrm{m}$, $\eta_{\mathrm{sampling}}$ increases to a peak at $d_p$ $12\,\mu\mathrm{m}$ and decreases to 0 % at $d_p \approx 40\,\mu\mathrm{m}$. The peak is introduced due to sub-isokinetic conditions, whereas the sharp decrease starting at $20\,\mu\mathrm{m}$ is mainly caused by gravitational losses in the horizontal part of the inlet and losses in the bend. However, it should be noted that for wind speeds $U_0 < 5\,\mathrm{m\,s^{-1}}$ and particles with $d_p > 30\,\mu\mathrm{m}$, most of the formulas are not valid.

## G4    Sigma-2 inlet of the CDA

The CDA used a Sigma-2 sampling head (Palas GmbH). The Association of German Engineers (VDI-2119, 2013) validated the Sigma-2 sampler after testing it in numerous investigations and concluded that it is a reliable collector for coarse and super-coarse particles (Dietze et al., 2006; Waza et al., 2019; Rausch et al., 2022; Tian et al., 2017). However, due to the low concentrations measured for particle diameters $d_p >\ 20\,\mu\mathrm{m}$ compared to the other instruments, we assume that substantial losses occurred for particles from that diameter on. We assume the inlet to have less gravitational losses than the directional inlet due to missing horizontal sampling lines. We also expect it to be able to sample from all horizontal angles, but to have losses due to the wire grid inside the head and the side panels between the three openings. To our knowledge, no inlet efficiency simulations or measurements were conducted and existing formulas cannot be adapted to the needs of the Sigma-2 head, so that we cannot quantify $\eta_{\mathrm{sampling}}$ for the Sigma-2 inlet. Conducting own numerical inlet simulations is beyond the scope of this paper, but we consider doing so in the future.





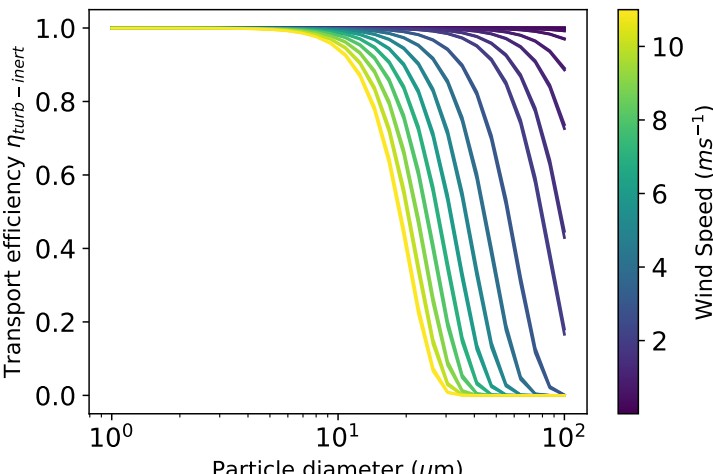

**Figure G2.** Turbulent inertial deposition within the UCASS for different wind conditions $U_0$ estimated based on Eq. G9.

## G5  UCASS

The UCASS are also a passive system, nearly open-path where the electronics are placed in a housing with a 2.2 cm (small side) wide oval opening. By using the same calculations as for the Welas and Fidas inlet and assuming approximately the same air flow inside and outside the instrument, we estimated minimal particle losses in the housing and a high gravitational efficiency close to one throughout the whole particles range. In the large opening of the UCASS, high Reynolds numbers can develop. When applying Eq. G9 to a simplified geometry of the UCASS with two parts (enlargement and straight pipe: L= 4.5 cm and 9 cm and D = 4.3 cm and 6.4 cm for the different parts), we found substantial losses due to turbulent inertial deposition as shown in Fig. G2. For higher wind speeds around 10 m s$^{-1}$, inlet efficiencies rapidly decrease from approximately 90% for 11 μm particles to nearly 0% for particles of 20 μm due to the large pipe diameters, high flow rates, and resulting high Reynolds numbers (Eq. G9). Based on the calculations explained above and especially due to turbulent inertial deposition, we would not expect to measure particles with diameters larger than $d_p = 20$ μm. However, as we measured these large particles, $\eta_{\text{tube, turb-inert}}$ might be overestimated by Eq. G9.

## G6  SANTRI2

Operating inlet-free, the SANTRI2 relies on the wind field to guide the particles through the optical path. Although this design avoids inlet-induced biases, potential turbulence effects caused by the underlying sensor platform can still lead to modifications of the aerosol sample seen by the sensors compared to its surrounding. Due to its slim design, we expect these alterations to be small.



## Appendix H: PSD for triaxial particle assumption

Although it would be more realistic to assume triaxial ellipsoids rather than biaxial ones for geometric diameters (Huang et al.,
2020), the lack of a suitable database for optical conversions of PSL particles with diameters $d > 50\,\mu m$ limits the applicability
of this approach, as the Meng et al. (2010) database does not cover such large particles. Figure H1 presents 15-minute average
size distributions of mass concentration for dusty conditions on 29 September 2022, focusing on three consecutive 15-minute
intervals from 14:30 until 15:00 UTC. The PSDs in Figure H1c show the corrected PSDs assuming triaxial ellipsoids and
after correction steps as explained in Appendix C-E are applied. They reveal differences compared to the biaxial assumption
in Figure H1b. Notably, the Welas size range is not entirely covered since the Meng et al. (2010) database does not include
particles from around $50\,\mu m$. Especially for Welas_4m, only particles $d_p < 70\,\mu m$ are covered whereas Welas_2m covers
also larger particles due to the diameter correction. Furthermore, SANTRI2 are asigned to a smaller size range, due to the
consideration of HWR=0.45, i.e., the third axis of the ellipsoid differing from the other two. Huang et al. (2021) assumes
$HWR = 0.4$ but this is the smallest value in the database of Meng et al. (2010). Additionally, the PSDs of Welas_2m show
a less smooth behavior likely due to the applied correction mechanisms. Furthermore, the peak of the size distributions shift
slightly towards smaller diameters, but its position is ambiguous between 30 and $50\,\mu m$.

These results demonstrate that the estimation of geometric diameters is highly sensitive to particle shape assumptions and
refractive index, the latter being estimated from Di Biagio et al. (2019) but not retrieved directly from the field data. As
the interpretation depends on these parameters, further research is essential to improve the reliability of geometric diameter
estimates, particularly for larger particles where uncertainties remain substantial.

*Data availability.* The data presented here will be available in a public repository upon acceptance of the manuscript.

*Author contributions.* HM processed aerosol spectrometer and most meteorological data, performed the analyses, created most figures,
and drafted the article. MK supervised the work with contributions from PK, CPG-P, and KK. KK conducted the FPS sampling, SEM
analysis, and provided the corresponding results. SD provided the sonic anemometer data from the $10\,m$ meteorological mast. JE provided
the transformation from optical to geometric diameter for the Fidas, CDA, and Welas. AG-R, KK, SD, MI, AA, XQ, TH, AW, FV, MK,
and CPG-P implemented the field campaign. JG and CS provided the bin classification for the UCASS and corresponding scientific and
technical support. OM contributed one UCASS and two Welas devices. VE and GN provided the four SANTRI2 and two SANTRI devices
and corresponding scientific and technical support. FW contributed the CDA and provided scientific and technical support for CDA, Welas,
and Fidas. CGF contributed to campaign preparation and provided support for the analysis. CPG-P and MK proposed and designed the
measurement campaign with contributions from AW, AA, XQ, SD, KK, and TH. MK calculated the flux density, created Figs. 1b,d, and
edited the manuscript. All authors provided feedback on the structure and/or the content of the final manuscript.







**Figure H1.** 15 minute average size distributions of mass concentration for dusty conditions on 29 September 2022 for 3 subsequent 15-min time periods from 14:30 until 15 UTC. (a) Uncorrected PSD with optical diameters (except SANTRI2, which uses projected-area diameter) and (b) corrected PSDs with geometric diameters assuming biaxial ellipsoids. (c) Same as (b) but with geometric diameters assuming triaxial ellipsoids. Standard errors are indicated by vertical lines (only positive errors are shown). Average 4 m friction velocity, $u_*$ for each 15-min period are indicated in the panel titles. Dashed lines indicate the size ranges of the dust size classifications (Adebiyi et al., 2023).



*Competing interests.* The authors declare that they have no conflict of interests

*Acknowledgements.* J-WADI was funded through the Helmholtz Association's Initiative and Networking Fund (grant no. VH-NG-1533), which primarily funded the work presented here, and the European Research Council under the Horizon 2020 research and innovation

programme through the ERC Consolidator Grant FRAGMENT (grant no. 773051). BSC co-authors also acknowledge support from the AXA Research Fund through the AXA Chair on Sand and Dust Storms at the Barcelona Supercomputing Center (BSC), and the Spanish Ministerio de Economía y Competitividad through the HEAVY project (grant PID2022-140365OB-I funded by MICIU/AEI/10.13039/50110001 and by ERDF, EU). Jerónimo Escribano acknowledges his AI4S fellowship within the "Generación D" initiative by Red.es, Ministerio para la Transformación Digital y de la Función Pública, for talent attraction (C005/24-ED CV1), funded by NextGenerationEU through PRTR. We

are grateful to T. Gamer, S. Haaß, B. Deny, S. Scheer, S. Vergara Palacio, M. Gonçalves Ageitos, L. Ilic, R. Sousse Villa, R. Miller, and A. Böhmländer for their support during campaign preparation and implementation. We acknowledge Wadi Rum Protected Area, Aqaba Special Economic Zone Authority, and the Directorate of Environmental Monitoring and Assessment at Ministry of Environment for their efforts to facilitate and organize the local infrastructure needed for the J-WADI. We acknowledge the use of ChatGPT (OpenAI) for assistance in refining the wording of certain sections of this manuscript.



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
