# Peer review of "From fine to giant: Multi-instrument assessment of the dust particle size distribution at an emission source during the J-WADI field campaign"

_EGUsphere, 2025_

## Referee Comment (RC1)

**Review of "From fine to giant: Multi-instrument assessment of the dust particle size distribution at an emission source during the J-WADI field campaign"**

The study sheds light on coarse dust emission PSD characteristics, which is extremely difficult to measure yet important for dust transport understanding and modelling, and impacts of dust on weather and climate. This is a thorough, interesting and novel study and I commend the authors for their attention to detail and impressive results. Overall I applaud the authors for clarity of method, data collection and processing.

The article reads well, despite being quite long. In a few limited places information seems to be missing, which may be a question of better signposting or inserting some clarifications. I suggest that discussion around the main findings could be much stronger. Other than the main points below, all comments are minor and relate to improved readability and signposting through the article.

Main points:

- The method applied to harmonize the PSDs appeared missing – i.e. how PSDs from multiple instruments (e.g. fig 9) were combined to give one single PSD (e.g. fig 11). This should be clearly explained and included in the main article as it is a very important part of the data analysis and processing.
- The article is weak on scientific discussion of the very interesting findings relating PSD to u* and stability. This is touched on in the discussion around figure 16, but could be much more extensive and thorough. How do the results agree/disagree with previous work relating PSD to wind speed? This has been an ongoing discussion in dust emission for decades and the authors are in a position to shed new light on this. See further comments below.
- Some sections require further clarification or explanation – see specific points below. In particular, information regarding the methodology of the FPS approach.
- It would be very insightful to provide a composite of diurnal averages of figure 8, since there is a clear diurnal cycle in weather variables, including wind speed and stability and therefore dust emission.
- The article would be enhanced by providing some statistics/data on number and surface area distributions, in addition to mass PSDs, since number/surface area affect various dust-climate interactions such as CCN, radiation etc. See further points below.

Specific points:

Abstract - I recommend adding something about the findings relating to atmospheric stability.

Can friction velocity mentioned in the abstract be provided along with windspeed(s)? This would provide a more relatable variable of wider application.

Line 11 – typo in symbol for friction velocity? (threshold friction velocity is given).

Introduction

Paragraph starting line 38 is not quite correct in terms of warming/radiative effects. Be clear on whether you mean TOA warming/cooling, or specifically atmospheric heating/cooling, which refers to the atmospheric column. All dust will cause an atmospheric SW heating, while it is the TOA effect whose sign can vary as a result of the strength of the atmospheric heating, which is sensitive to particle size. In the LW, dust cools the atmosphere but causes a positive radiative

effect at the TOA. Please correct and sharpen the scientific terminology here. Be specific on whether you mean shortwave, longwave or total radiation. If SSAs are mentioned, wavelengths should be specified.

L44 – unclear what you are saying causes the decrease in SSA from 1 to 0.95?

L107 – if the site is downwind of a solar plant, does this have any associated local particulate emissions which could influence measurements?

Sect 2.1 – how were the instruments powered?

Fig 1 caption – (c) – 'instruments used' – add 'in this article' – it is not entirely clear what the pink/black labelled instruments are – please make this clear.

L122 and next few lines – add a definition of the meanin of Obukhov length, and how u* and L were obtained from the scintillometer.

L137 – it's not clear which markers on the figure relate to the different masts mentioned.

Two SANTRI instruments are used – SANTRI and SANTRI2. I was somewhat confused as I read the threshold friction velocity section (~L145). Why is SANTRI not included in the instrument table? It would probably be useful to have a table of meteorological instruments/variables at this stage in the paper to help navigation. Further confusion – the map on fig 1 shows a 'SANTRI4.'

Table 1 caption – add that diameters are optical/projected diameter.

L204 – 'accurately' model is a bit misleading – it accurately applies Mie theory, though this in itself has various limitations around dust particle measurement.

L208 – Table 1 gives a different max d for the Fidas

L239 – was particle density measured during the campaign?

L247 – what does 'finer' than the emission flux PSD mean specifically? More resolution in diameter space?

L260 – 'primarily' – perhaps 'only' – as only asphericity and RI were considered?

L277 (&C2, UCASS) – this is an interesting approach – has this been previously applied in other studies, and would it be relevant to other scattering instruments for PSD?

L338 – NDR as an acronym is not expanded

L338-360 it is unclear what the motivation for these paragraphs is. A bit of background on the FPS system and how it can be used is required, or a statement to this effect. I'm guessing the measurements from the SEM are being combined with deposition assumptions, to recreate an atmospheric airborne PSD? Why can the sampled data not simply be used to represent atmospheric concentrations? Please make this clearer for the benefit of those not familiar with FPS sampling.

L361-366 – section numbers don't quite match up with what is presented in each section.

Fig 8 – it would be more natural to give mass concentration units in milligrams or micrograms per cubic meter, and avoid many decimal places.

L368 and Fig 8 – are temperatures dry bulb temperature or potential temperature? The symbol theta should only be used for potential temperature, T for standard dry bulb.

Fig 8 – this figure contains a wealth of information which is skipped over quite quickly. It would be interesting to see a daily composite version of the figure, given the strong diurnal cycle. This would also emphasize the times of day which certain features discussed occur at. This could also shed some novel light on dust emission processes and how they are affected by the diurnal cycle, which impacts many areas, including satellite detection by overpasses at certain times, and model emission processes.

Fig 8 caption – include what the dashed line in f and g represents.

Whole article – It is confusing that results are presented and described in UTC, rather than local time. Given the importance of the diurnal cycle, local time would be much more informative and avoid confusion. I suggest switching to local time, if possible.

Section 3.1 – there appears to be a change in regime between around the start of the campaign to around 22 Sept, and then 22 Sept to 30 Sept or possibly the end of the campaign. Meteorology switches from cool/humid/low pressure to warm/dry/higher pressure, and it looks like the behaviour of the diurnal cycle in wind changes too. Has this been investigated and what caused it? Was there any impact on dust measurement results?

Fig 9 caption – would be useful to add 'fine/coarse/super-coarse/giant' after classifications.

Fig 9 – top row – UCASS A appears to have no error bars?

L408 – seems like the roughly constant concentration starts at more like 5 microns rather than 10

L411 – 'both instruments maintain similar distribution shapes' – I would disagree – they look fairly different at d>10um in fig 9a.

L419-420 – this statement is vague and not very helpful – can you be more specific about the UCASS instruments? Also make it clear that both are excluded from further analysis.

L425 – why would the CDA have a lower sensitivity in this size range? Inlet efficiency seems logical, but can this be explained based on the instrument design/engineering?

P21 'use of inlets' section – first paragraph – please can you re-state whether the inlet sampling efficiency has been applied to the data in figure 9.

L518 – it would be helpful to add a sentence summarizing the main finding of this part of the appendix.

Section 3.5/Fig 11 onwards – The article seems to suddenly skip to using harmonized PSDs from the SANTRI2/Welas/Fidas, without explaining how this is achieved, since in some size ranges the instruments duplicate data, while in others there are measurement gaps, and in some instances the SANTRI2 does not follow well from the other instruments. What processing has been done here? This deserves its own subsection to explain. I was surprised by this given the attention to detail thus fur, and went back to search for something I missed but could not find anything.

Fig 11 – it is somewhat difficult to pick out the individual colors of the legend – can the legend color lines be made thicker? It would also be useful to add diameter tick marks to the top axis.

L553-555 – it is interesting that the PSD starts to respond at around d=60 microns in this u* range, but not at smaller particle sizes, and might be worth mentioning.

L550 – although the large variability here detracts from this finding.

L561 – should be 'as u* increases'?

Fig 12 – it would be interesting to consider adding bars to show the same % contributions, but for surface area and number, as these parameters contribute to different processes (e.g. radiative effects, CCN/IN, Mahowald et al. (2014)). Three bars for each u* range could be displayed, instead of just 1 corresponding to mass.

L581-2 – or more likely due to the large errors (variability) for this size range which we see in fig 11, which cannot be shown on fig 12.

L583-587 and fig 13 discussion – relating to the discussion here it would be useful to have a version of fig 8 composited into a diurnal cycle, since the stability and therefore the PSD are so dependent on time of day.

L592 – 'No clear trend…' – only in the normalized figure (b). The trend in fig 13a seems clear for the largest sizes.

Fig 14 – the definitions and ranges on L130 exceed the values shown on the color bar in the figure (e.g. unstable and stable are not included). Given this some of the discussion around the different types of stability in the figure are a bit difficult to follow. Ideally it would be good to make the color bar blocked, with different colours corresponding to specific stability classes. Specifically, 'For instance, for u∗ = 0.25m s−1 and for near unstable conditions, mass concentrations could reach approximately one order of magnitude higher than the average for unstable conditions' – the data doesn't seem to suggest this.

L613 – what were the findings of these studies?

L623/fig 15 – would be useful to state why we now see PSDs in deposition rates rather than concentrations.

Fig 15 – why no error bars for the FPS?

Fig 15 – the data is quite small and possibly makes discrepancies look better. I suggest at least adding additional tick marks for dDM every order of 10 magnitude to make interpretation easier.

L638-9 – in a different field experiment? Same instrumentation?

L644 paragraph – it would be useful to recap on what size the deposition velocity used represents.

Fig 16 – can the normalization process be made clearer? It is also unclear what 'MEAN' refers to – this needs to be made clearer in the caption, legend and main text. 'SOURCE' data should be added to the caption.

L601 – avoid 'elevated' dust conditions which might be interpreted as altitude.

Discussion around the main findings: L695 – how does the J-WADI compare to the other listed data sources in fig 16? Were they ground-based and at emission sources? Would we expect similar PSDs? Were they limited in maximum diameter measured? A discussion on similarities and differences, and causes for these between campaigns, should be included, and related

back to paragraph L78-83 in the introduction. Since you find that u* influences PSD, does this contradict the findings of Kok (2011) and the move of some models to implement this emission scheme?

L712 – friction velocity or threshold friction velocity?

L712 – 'during *active* dust emission events' – I was under the impression that the PSDs presented had been campaign averages, with no filtering based on 'active' periods or other – please clarify.

L718-723 – it would be useful to revisit the smaller super-coarse/giant concentrations and PSDs under unstable conditions here, with an explanation for this.

It would be entirely acceptable to move the extensive appendices to a supplement, should the authors wish to reduce the length of the main manuscript.

---

## Author Comment (AC1)

**Response to Reviewer #1**

We thank Reviewer #1 for their thorough and constructive review of our manuscript *"From fine to giant: Multi-instrument assessment of the dust particle size distribution at an emission source during the J-WADI field campaign"*. We appreciate the recognition of the novelty and detail of our work, and we have considered all comments carefully. Below, we provide point-by-point responses. Reviewer comments are italicized, followed by our responses.
* * *
*Main points:*

**1.** *Harmonization of PSDs (Fig. 9 to Fig. 11) is not clearly explained.*

*(The method applied to harmonize the PSDs appeared missing – i.e. how PSDs from multiple instruments (e.g. fig 9) were combined to give one single PSD (e.g. fig 11). This should be clearly explained and included in the main article as it is a very important part of the data analysis and processing.)*

> We agree that the harmonization process is a crucial part of our methodology. We have now included a link to the appendix where we explained how the PSDs from SANTRI2, Welas, and Fidas were combined, including how overlapping size ranges and measurement gaps were handled.

**2.** *Discussion relating PSD to $u_*$ and stability could be stronger.*

*(The article is weak on scientific discussion of the very interesting findings relating PSD to u\* and stability. This is touched on in the discussion around figure 16, but could be much more extensive and thorough. How do the results agree/disagree with previous work relating PSD to wind speed? This has been an ongoing discussion in dust emission for decades and the authors are in a position to shed new light on this. See further comments below.)*

> We have extended the discussion in Section 3.5 and added further contextualization regarding existing literature. In particular, we now explicitly contrast our findings with Kok (2011), Shao et al. (2020), and Khalfallah et al. (2020), and discuss dependencies of the PSD on both friction velocity and atmospheric stability based on our data.

**3.** *Clarification of FPS methodology is needed.*

*(Some sections require further clarification or explanation – see specific points below. In particular, information regarding the methodology of the FPS approach.)*

> We have expanded the relevant section to provide clearer background on the FPS approach and explicitly describe the assumptions involved in using deposition measurements to infer airborne PSDs. This includes explaining why direct representation of concentration PSD from sampled data is not feasible.

**4.** *Suggestion to add a diurnal composite version of Fig. 8.*

*(It would be very insightful to provide a composite of diurnal averages of figure 8, since there is a clear diurnal cycle in weather variables, including wind speed and stability and therefore dust emission.)*

> We have created and added a figure showing the diurnal average of Fig. 8.

**5.** *Include number and surface area PSDs.*

*(The article would be enhanced by providing some statistics/data on number and surface area distributions, in addition to mass PSDs, since number/surface area affect various dust-climate interactions such as CCN, radiation etc. See further points below.)*

> We chose to focus on mass PSDs in the main text. However, examples of number distributions are now included in the Supplementary Material.
* * *
**Specific points:**

**Abstract:**

> *(I recommend adding something about the findings relating to atmospheric stability.)*

- We now mention atmospheric stability in the abstract.

  *(Can friction velocity mentioned in the abstract be provided along with windspeed(s)? This would provide a more relatable variable of wider application)*

- We also added the threshold wind speed at 4m height of $3.3\,\mathrm{m\,s^{-1}}$.

  *(Line 11 – typo in symbol for friction velocity? (threshold friction velocity is given).)*

- The friction velocity symbol has been corrected.

**Line 38 (Radiative effects):**

(Paragraph starting line 38 is not quite correct in terms of warming/radiative effects. Be clear on whether you mean TOA warming/cooling, or specifically atmospheric heating/cooling, which refers to the atmospheric column. All dust will cause an atmospheric SW heating, while it is the TOA effect whose sign can vary as a result of the strength of the atmospheric heating, which is sensitive to particle size. In the LW, dust cools the atmosphere but causes a positive radiative effect at the TOA. Please correct and sharpen the scientific terminology here. Be specific on whether you mean shortwave, longwave or total radiation. If SSAs are mentioned, wavelengths should be specified.)

> We corrected and sharpened the terminology: TOA effect can vary in sign with size of particles. We also clarify the roles of shortwave (SW), longwave (LW), and total radiation when discussing SSA.

**Line 44 (SSA decrease):**

*(L44 – unclear what you are saying causes the decrease in SSA from 1 to 0.95?)*

> The sentence now clarifies that the decrease in SSA from 1 to 0.95 results from increasing particle size.

**Line 107 (solar plant influence):**

*(L107 – if the site is downwind of a solar plant, does this have any associated local particulate emissions which could influence measurements?)*

>We added that the distance between the measurement site and the solar plant is large enough that we do not expect any influence on local dust emissions.

**Section 2.1 (Power supply):**

*(Sect 2.1 – how were the instruments powered?)*

>Added explanation of how all instruments were powered during the campaign.

**Fig. 1 caption:**

*(Fig 1 caption – (c) – 'instruments used' – add 'in this article' – it is not entirely clear what the pink/black labelled instruments are – please make this clear.)*

>Clarified that instruments marked in pink are those used *in this article*; we also added corresponding abbreviations.

**Lines 122ff (Obukhov length, u*\*, L):**

*(L122 and next few lines – add a definition of the meanin of Obukhov length, and how u\* and L were obtained from the scintillometer.)*

>Added definitions of Obukhov length and descriptions of how u\* and L were derived from the scintillometer data.

**Marker confusion in Fig. 1 and SANTRI instruments:**

*(L137 – it's not clear which markers on the figure relate to the different masts mentioned.*

*Two SANTRI instruments are used – SANTRI and SANTRI2. I was somewhat confused as I read the threshold friction velocity section (~L145). Why is SANTRI not included in the instrument table? It would probably be useful to have a table of meteorological instruments/variables at this stage in the paper to help navigation. Further confusion – the map on fig 1 shows a 'SANTRI4.')*

>Added a table listing all meteorological instruments and clarified that SANTRI measures saltation, not aerosol. The label in the figure was changed to "aerosol spectrometers" for clarity. SANTRI instead of SANTRI4 was labeled in the figure.

**Table 1 caption:**

*(Table 1 caption – add that diameters are optical/projected diameter.)*

>We added that diameters are optical/projected diameters.

**Lines 204-208 (Mie theory, Fidas diameters):**

*(L204 – 'accurately' model is a bit misleading – it accurately applies Mie theory, though this in itself has various limitations around dust particle measurement.*

*L208 – Table 1 gives a different max d for the Fidas)*

>Changed wording to reflect that Mie theory is applied with known limitations.

**Line 239 (Particle density):**

*(L239 – was particle density measured during the campaign?)*

> We clarified that density was not measured in the field; assumed values are based on literature.

**Line 247 ("Finer" resolution):**

*(L247 – what does 'finer' than the emission flux PSD mean specifically? More resolution in diameter space?)*

> Clarified that "finer" refers to higher concentrations in smaller diameters.

**Line 260 ("Primarily" vs. "Only"):**

*(L260 – 'primarily' – perhaps 'only' – as only asphericity and RI were considered?)*

> We changed "primarily" to "only," since only asphericity and refractive index were considered.

**UCASS correction (L277):**

*(L277 (&C2, UCASS) – this is an interesting approach – has this been previously applied in other studies, and would it be relevant to other scattering instruments for PSD?)*

> Clarified that the correction method was newly developed based on the outlier behavior of UCASS and has not yet been applied to other instruments.

**Line 338 (NDR):**

*(L338 – NDR as an acronym is not expanded)*

> NDR acronym is now expanded.

**Lines 338–360 (FPS clarification):**

*(L338-360 it is unclear what the motivation for these paragraphs is. A bit of background on the FPS system and how it can be used is required, or a statement to this effect. I'm guessing the measurements from the SEM are being combined with deposition assumptions, to recreate an atmospheric airborne PSD? Why can the sampled data not simply be used to represent atmospheric concentrations? Please make this clearer for the benefit of those not familiar with FPS sampling.)*

> Improved the description to better explain the motivation and methodology of FPS analysis.

**L. 361-366:**

*(L361-366 – section numbers don't quite match up with what is presented in each section.)*

> Matched section numbers.

**Fig. 8 (units and interpretation):**

*(Fig 8 – it would be more natural to give mass concentration units in milligrams or micrograms per cubic meter, and avoid many decimal places.*

*Fig 8 caption – include what the dashed line in f and g represents.*

*Fig 8 – this figure contains a wealth of information which is skipped over quite quickly. It would be interesting to see a daily composite version of the figure, given the strong diurnal cycle. This would also emphasize the times of day which certain features discussed occur at. This could also shed some novel light on dust emission processes and how they are affected by the diurnal cycle, which impacts many areas, including satellite detection by overpasses at certain times, and model emission processes.)*

> Concentrations were changed to scientific notation for better readability. Dashed lines in panels f and g are now explained in the caption. We added a diurnal cycle to the supplementary material.

**L. 368 and Fig. 8, Temperature variable:**

*(L368 and Fig 8 – are temperatures dry bulb temperature or potential temperature? The symbol theta should only be used for potential temperature, T for standard dry bulb.)*

> Clarified that temperature is dry bulb temperature, and theta notation was revised.

**UTC vs. Local Time:**

*(Whole article – It is confusing that results are presented and described in UTC, rather than local time. Given the importance of the diurnal cycle, local time would be much more informative and avoid confusion. I suggest switching to local time, if possible.)*

> The conversion to local time is indicated in the text; for consistency with other studies, we keep the UTC timestamps in the figures.

**Campaign regime shift (Sect. 3.1):**

*(Section 3.1 – there appears to be a change in regime between around the start of the campaign to around 22 Sept, and then 22 Sept to 30 Sept or possibly the end of the campaign.*

*Meteorology switches from cool/humid/low pressure to warm/dry/higher pressure, and it looks like the behaviour of the diurnal cycle in wind changes too. Has this been investigated and what caused it? Was there any impact on dust measurement results?)*

> We acknowledge that a shift in meteorological regime occurred but did not further investigate this due to the scope of the current study.

**L. 408:**

*(L408 – seems like the roughly constant concentration starts at more like 5 microns rather than 10)*

> We adjusted the description to 5-10 um.

**L. 411:**

*(L411 – 'both instruments maintain similar distribution shapes' – I would disagree – they look fairly different at d>10um in fig 9a.)*

> We changed the description according to your comment. The peaks are at similar diameters and therefore the shapes are similar even though at different concentrations. We removed the subsentence to avoid confusion.

**L 419-420:**

*(L419-420 – this statement is vague and not very helpful – can you be more specific about the UCASS instruments? Also make it clear that both are excluded from further analysis.)*

W**e** changed to: "The oscillations of the PSD, i.e., the classification of size bins and the conversion from scattering cross section to particle diameter, seems to be unrealistic in comparison to the other instruments, and further adjustments may be needed to optimize particle categorization and UCASS A and B are therefore excluded from further analysis."

**L 425:**

*(L425 – why would the CDA have a lower sensitivity in this size range? Inlet efficiency seems logical, but can this be explained based on the instrument design/engineering?)*

Explained why the CDA might show lower sensitivity due to design aspects (e.g., light source and sampling volume).

**P21:**

*(P21 'use of inlets' section – first paragraph – please can you re-state whether the inlet sampling efficiency has been applied to the data in figure 9.)*

added that it is applied to data shown in Fig. 9b

**Fig. 9 and related lines:**

*(Fig 9 caption – would be useful to add 'fine/coarse/super-coarse/giant' after classifications.*

*Fig 9 – top row – UCASS A appears to have no error bars?)*

Added fine/coarse/super-coarse/giant labels in the caption. UCASS A error bars are now shown. Clarified discrepancies in distributions and removed vague statements.

**Line 518 (Appendix summary):**

*(L518 – it would be helpful to add a sentence summarizing the main finding of this part of the appendix.)*

A summary sentence was added.

**Harmonization:**

*(Section 3.5/Fig 11 onwards – The article seems to suddenly skip to using harmonized PSDs from the SANTRI2/Welas/Fidas, without explaining how this is achieved, since in some size ranges the instruments duplicate data, while in others there are measurement gaps, and in some instances the SANTRI2 does not follow well from the other instruments. What processing has been done here? This deserves its own subsection to explain. I was surprised by this given the attention to detail thus fur, and went back to search for something I missed but could not find anything.)*

Harmonization procedure is explained in the appendix and now properly linked.

**Fig. 11:**

*(Fig 11 – it is somewhat difficult to pick out the individual colors of the legend – can the legend color lines be made thicker? It would also be useful to add diameter tick marks to the top axis.)*

Legend lines were thickened. Diameter ticks were added to top axis.

**L553-555:**

*(L553-555 – it is interesting that the PSD starts to respond at around d=60 microns in this u\* range, but not at smaller particle sizes, and might be worth mentioning.)*

Added that the peak is the same as for larger u\*.

**L550:**

*(L550 – although the large variability here detracts from this finding.)*

Added that: although the level of increase varies.

**L561:**

*(L561 – should be 'as u\* increases'?)*

No, for smaller u\*, there is not much of a pattern for increasing u\* leading to increasing concentrations To make this clearer we changed "as u\* increases" to "for small u\* values"

**Fig 12:**

*(Fig 12 – it would be interesting to consider adding bars to show the same % contributions, but for surface area and number, as these parameters contribute to different processes (e.g. radiative effects, CCN/IN, Mahowald et al. (2014)). Three bars for each u\* range could be displayed, instead of just 1 corresponding to mass.)*

We added examples to the supplementary material.

**L581f:**

*(L581-2 – or more likely due to the large errors (variability) for this size range which we see in fig 11, which cannot be shown on fig 12.)*

We added a subsentence for clarity: "This ambiguous dependency suggests that the large variability for this size range which can be observed in Fig. \ref{fig:u_star} (not shown in Fig. \ref{fig:u_star_scatter}) and low statistics blur the trend or factors other than friction velocity alone may influence the concentration of larger particles in the PSD. "

*L583ff, Fig. 13:*

*(L583-587 and fig 13 discussion – relating to the discussion here it would be useful to have a version of fig 8 composited into a diurnal cycle, since the stability and therefore the PSD are so dependent on time of day.)*

*We added a figure to the supplementary material.*

**L592:**

*(L592 – 'No clear trend...' – only in the normalized figure (b). The trend in fig 13a seems clear for the largest sizes.)*

We added Fig. 13b

**Fig. 14 colorbar:**

*(Fig 14 – the definitions and ranges on L130 exceed the values shown on the color bar in the figure (e.g. unstable and stable are not included). Given this some of the discussion around the different types of*

*stability in the figure are a bit difficult to follow. Ideally it would be good to make the color bar blocked, with different colours corresponding to specific stability classes.*

*Specifically, 'For instance, for u∗ = 0.25m s−1 and for near unstable conditions, mass concentrations could reach approximately one order of magnitude higher than the average for unstable conditions' – the data doesn't seem to suggest this.)*

> We prefer a continuous colorbar due to smooth transitions between stability regimes, but we have adjusted the legend to better align with the textual definitions of stability ranges. The sentence about near-unstable conditions was revised for clarity.

**L. 613:**

*(L613 – what were the findings of these studies?)*

> Added findings of these studies.

**L. 613:**

*(L613 – what were the findings of these studies?)*

> Added findings of these studies.

**Fig. 15 and FPS (e.g., deposition velocity, dDM ticks, instrument differences):**

*(L623/fig 15 – would be useful to state why we now see PSDs in deposition rates rather than concentrations.*

*Fig 15 – why no error bars for the FPS?*

*Fig 15 – the data is quite small and possibly makes discrepancies look better. I suggest at least adding additional tick marks for dDM every order of 10 magnitude to make interpretation easier.*

*L638-9 – in a different field experiment? Same instrumentation?*

*L644 paragraph – it would be useful to recap on what size the deposition velocity used*

*represents.)*

> All these points have been addressed as suggested. Added: "to make them comparable with the FPS samples." and error bars for FPS. Tick marks were added for dDM. Clarified the size range represented by deposition velocity. Discussed differences between field experiments and instrumentation. Where appropriate, references to the supplemental material are provided. Deposition velocity was added.

**Fig. 16:**

*(Fig 16 – can the normalization process be made clearer? It is also unclear what 'MEAN' refers to – this needs to be made clearer in the caption, legend and main text. 'SOURCE' data should be added to the caption.)*

> Normalization methods and labels (e.g., "MEAN", "SOURCE") are clarified in the figure caption and main text.

**L. 601:**

*(L601 – avoid 'elevated' dust conditions which might be interpreted as altitude.)*

changed elevated to increased.

**Discussion around the main findings:**

*(Discussion around the main findings: L695 – how does the J-WADI compare to the other listed data sources in fig 16? Were they ground-based and at emission sources? Would we expect similar PSDs? Were they limited in maximum diameter measured? A discussion on similarities and differences, and causes for these between campaigns, should be included, and related back to paragraph L78-83 in the introduction. Since you find that u\* influences PSD, does this contradict the findings of Kok (2011) and the move of some models to implement this emission scheme?)*

> Included a discussion about the differences between the different campaigns and what to expect

**Friction velocity, active events, and conclusion points:**

*(L712 – friction velocity or threshold friction velocity?*

*L712 – 'during active dust emission events' – I was under the impression that the PSDs presented had been campaign averages, with no filtering based on 'active' periods or other – please clarify.*

*L718-723 – it would be useful to revisit the smaller super-coarse/giant concentrations and PSDs*

*under unstable conditions here, with an explanation for this.)*

> Line 712 refers to friction velocity (corrected). We clarified that the PSDs refer to active dust events. The connection to stability and u\* was further elaborated in Section 3.5, although it is not reiterated in the conclusion due to the complexity of the stability–u\* relationship.

**Suggestion to move appendix to supplement:**

*(It would be entirely acceptable to move the extensive appendices to a supplement, should the authors wish to reduce the length of the main manuscript.)*

> We appreciate the suggestion to move the Appendices to the Supplement. While we maintain them in the main manuscript for transparency and methodological completeness, we remain open to the editor's preference for restructuring if required.

We again thank the reviewer for their detailed and insightful comments, which helped us improve the clarity, depth, and rigor of the manuscript.

---

## Author Comment (AC2)

**Response to Reviewer #2**

We thank the reviewer for their positive assessment of our study and for highlighting the value of our dataset and experimental design. We acknowledge that the process of combining the PSDs from the different instruments was not sufficiently explained in the original manuscript.

Below, we provide a response. Reviewer comments are italicized, followed by our responses.
* * *
*The only thing I found odd is that I couldn't tell how the authors combined the PSDs for multiple instruments into the final PSD presented in figure 11. A couple of lines at the start of section 3.5 would probably be sufficient.*

We have now included a link to the appendix where it is explained how the final PSD presented in Fig. 11 from SANTRI2, Welas, and Fidas were combined, including how overlapping size ranges and measurement gaps were handled.